# Linear Bandits beyond Inner Product Spaces, the case of Bandit Optimal Transport

**Lorenzo Croissant** [1]

## Abstract

The general learnability of linear bandits has long been established when the objective is to minimise the inner product between a cost parameter and the decision variable. While this is highly general, reliance on an inner product structure belies the name of *linear* bandits, and fails to account for problems such as Optimal Transport. Using the Kantorovich formulation of Optimal Transport as an example, we show that an inner product structure is *not* necessary to achieve efficient learning in linear bandits. Proposing a refinement of the classical OFUL algorithm, using least-squares after embedding in a Hilbertian subspace, and constraining actions to this subspace through penalisation of optimism, we show that the trajectorial regret bounds of OFUL can be generalised. Using results from entropic transport and functional regression, we derive worst-case regret bounds interpolating between $\tilde{\mathcal{O}}(\sqrt{T})$ and $\mathcal{O}(T)$ depending on the regularity of the cost function, which recover the parametric rate $\tilde{\mathcal{O}}(d\sqrt{T})$ in finite-dimensional settings.

## 1. Introduction

Stochastic bandits are a class of sequential decision-making problems under uncertainty in which the agent aims to, say, minimise an unknown function $J^* : \mathbb{A} \to \mathbb{R}$ sequentially, with access only to a hypothesis class $\{J^\varsigma : \varsigma \in \Theta\}$, for some index set $\Theta$, and sequentially revealed noisy feedback, of the form $C_t = J^*(a_t) + \xi_t$, to the taken actions $(a_t)_{t \in \mathbb{N}} \subset \mathbb{A}$, where $(\xi_t)_{t \in \mathbb{N}}$ is a martingale difference sequence. In this paper, we will focus on the *realisable* case, in which there is $\varsigma^* \in \Theta$ such that $J^* = J^{\varsigma^*}$.

A bandit learning algorithm is defined as an $\mathbb{A}$-valued stochastic process $\boldsymbol{\pi} := (a_t)_{t \in \mathbb{N}}$ predictable w.r.t. the filtration of $(\xi_t)_{t \in \mathbb{N}}$ and is evaluated online using the *regret*, defined as the stochastic process $(\mathscr{R}_T^{J^*}(\boldsymbol{\pi}))_{T \in \mathbb{N}}$, with

$$\mathscr{R}_T^{J^*}(\boldsymbol{\pi}) := \sum_{t=1}^{T} C_t - \min_{a \in \mathbb{A}} J^*(a), \text{ for } T \in \mathbb{N}. \quad (1)$$

By evaluating $\boldsymbol{\pi}$ directly in terms of the objective $J^*$, regret minimisation forces algorithms' actions $(a_t)_{t \in \mathbb{N}}$ to trade-off *exploration* (taking statistically informative actions about $J^*$) and *exploitation* (taking known low-cost actions).

Because of the exploration-exploitation trade-off, the ability to extrapolate information about minimisers of $J^*$ from samples determines the hardness of a problem instance, and, thus, the regularity of $J^*$ is a key factor in achievable learning bounds.

In early historical examples (Lai & Robbins, 1985; Thompson, 1933; Auer et al., 2002), only finite numbers of actions were considered. By reducing the problem to a small set of credible hypotheses to test, these *multi-armed* bandit problems can achieve low-regret even on complicated classes of functions (Tran-Thanh & Yu, 2014).

For continuous action spaces, classes of Lipschitz (or Hölder) objectives have been studied at length, see, e.g., Bubeck et al. (2011); Magureanu et al. (2014); Kleinberg et al. (2019). The resulting bounds are of order[1] $\tilde{\mathcal{O}}(T^{(d+1)/(d+2)})$ when $\mathbb{A}$ is, effectively, a $d$-dimensional metric space (Kleinberg et al., 2019) (and this is tight up to logarithmic factors).

When $\mathbb{A}$ is infinite-dimensional, in contrast, results are known only when the action and hypothesis spaces are derived from a bilinear functional on a Hilbert space, i.e. of the form

$$J^\varsigma : a \in \mathbb{A} \mapsto \langle a | \varsigma \rangle_{\mathcal{H}}, \quad (2)$$

for a Hilbert space $(\mathcal{H}, \langle \cdot | \cdot \rangle_{\mathcal{H}})$. These *linear* bandits were introduced (with $\mathcal{H}$ finite-dimensional) by Auer (2003), and refined subsequently by Abeille & Lazaric (2017); Vernade et al. (2020); Hao et al. (2020), amongst others. In his

[1]CREST, ENSAE, & INRIA FairPlay team, Palaiseau, France. Correspondence to: Lorenzo Croissant <lorenzo.croissant@ensae.fr>.

*Proceedings of the 43rd International Conference on Machine Learning*, Seoul, South Korea. PMLR 306, 2026. Copyright 2026 by the author(s).

[1]Throughout, $\mathcal{O}$ denotes Landau notation and $\tilde{\mathcal{O}}$ hides logarithmic factors.

doctoral thesis, Abbasi-Yadkori (2012) provides a formal generalisation of the key concentration arguments, which holds in separable Hilbert spaces of arbitrary dimension. This *optimistic* algorithm, which we denote $\pi^\circ$, relies on the construction of Regularised Least Squares (RLS) estimators to obtain confidence sets $(\mathcal{C}_t)_{t \in \mathbb{N}} \subset \mathcal{H}$ for the unknown $\varsigma^*$, and then chooses the optimistic action

$$a_t \in \underset{a \in \mathbb{A}}{\operatorname{argmin}} \ \underset{\varsigma \in \mathcal{C}_t}{\inf} \ \langle a | \varsigma \rangle_{\mathcal{H}} \,,$$

at each time step $t \in \mathbb{N}$. The resulting $(1 - \delta)$-probability regret bound (Abbasi-Yadkori et al., 2011, Thm. 4.1) is

$$\mathscr{R}_T^{J^*}(\pi^\circ) \leq C \sqrt{T \left( \log \det \left( \operatorname{Id} + M_T M_T^* \right) + \log \left( \frac{1}{\delta} \right) \right)} \,,$$

wherein $M_T : \varsigma \in \mathcal{H} \mapsto (\langle a_t | \varsigma \rangle_{\mathcal{H}})_{t=1}^T$, $M_T^*$ is its adjoint, and $C > 0$. These bounds are *trajectorial*, i.e. they depend on the points actually observed (via $M_T M_T^*$).

To make regret bounds comparable across algorithms, it is standard to convert them into *worst-case* bounds, i.e. independent of the trajectory $(a_t)_{t \in \mathbb{N}}$, by bounding the log-determinant term. This is done through structural assumptions on the hypothesis space $\Theta$. In finite dimension, the workhorse is the famous elliptical potential lemma (see e.g. Lemma E.3 in Abbasi-Yadkori et al., 2011). When the dimension of $\mathcal{H}$ is countably infinite, one can rely on Reproducing Kernel Hilbert Space (RKHS) theory, as in Chowdhury & Gopalan (2017); Janz et al. (2020); Takemori & Sato (2021); and in Valko et al. (2013), who introduced a kernelised version of the above optimistic algorithm.

While the statistical properties of worst-case analysis are not inherently dependent on a Hilbertian structure, the trajectorial analysis decidedly is. In fact, the Cauchy-Schwarz inequality is an integral part of the construction of the confidence sets (see Abbasi-Yadkori et al., 2011, p. 23) as well as the control of the regret by their width (see Thm. 3.11 in Abbasi-Yadkori et al., 2011). This observation reveals a major conceptual gap in the literature on linear bandits:

> *what happens when $J^*$ is linear, but not an inner product in a Hilbert space?*

The name "linear bandits" suggests that linearity is the key property, but it is unclear that linearity itself is actually a sufficient condition to achieve sub-linear worst-case regret or even trajectorial bounds in infinite dimension. This paper aims to begin filling in this awkward gap in the terminology.

This gap is far from inconsequential: there are well-studied problems with practical applications which fit exactly into this description, notably integral functionals where either the actions or the "parameters" $\varsigma$ are measures. As a stepping stone to the general case, we will focus in this paper on the *Kantorovich Optimal Transport* problem, which is a linear functional on the space of probability measures. In this problem, permissible actions $\pi$ live in a specific subspace of the space of probability measures over a space $\mathcal{X}$ (detailed below), while $\varsigma$ is a function and the objective is given by

$$J^\varsigma(\pi) := \int \varsigma \, d\pi \,.$$

This problem is a well-studied problem in optimal transport theory (Villani, 2009), with numerous applications (Galichon, 2021). Importantly, it exhibits strong regularity properties which are independent of the dimension of the parameter space $\Theta$, which makes it a good candidate to study the infinite-dimensional learning problem. Our results confirm this intuition.

## 1.1. Contributions

At a high-level, we show that the Kantorovich problem is learnable in the sense of linear bandits for a wide range of cost functions $\varsigma$, and that the resulting regret bounds are of the same type as those of the optimistic algorithm of Abbasi-Yadkori et al. (2011) for linear bandits in Hilbert spaces. This shows that an inner product structure is *not* necessary to achieve the same type of regret bounds, though the question of sufficiency remains open for future work. From a technical perspective we make three contributions.

**1)** We show that the Kantorovich problem admits the same type of trajectorial regret bounds as the optimistic algorithm of Abbasi-Yadkori et al. (2011) (see Theorem 4.1). This requires modifying the Optimism in the Face of Uncertainty (OFU) paradigm to reduce the action space to a frequency-domain (Fourier) representation in which confidence sets can be constructed, and then to leverage the regularity properties of the Kantorovich problem to control the corresponding approximation error.

**2)** We show that this methodology generalises to any bilinear functional provided we can use a barrier functional to regularise the problem onto a Hilbertian subspace. This resulting approximation term in the regret is then controlled by the rate of convergence from the regularised problem to the original one, e.g. via $\Gamma$-convergence (see Theorem D.3).

**3)** Finally, we show that the Kantorovich problem is bandit-learnable by deriving worst-case regret bounds from the trajectorial ones, which interpolate between $\tilde{\mathcal{O}}(\sqrt{T})$ and $\tilde{\mathcal{O}}(T)$ depending on the regularity of the cost function. This is achieved by combining the optimistic algorithm with functional regression techniques, whose statistical efficiency can be tied to the decay rate of the Fourier transform of the cost function in a relevant basis. In particular, we recover the tight rates $\mathcal{O}(p\sqrt{T})$ for parametric problems in dimension $p$ (up to logarithmic factors) and show a rate of the form

$\tilde{\mathcal{O}}(T^{\frac{q+4}{2q+4}})$ for non-parametric problems with a decay rate of the Fourier transform of the cost function of $t^{-q}$ for $q > 0$. This applies, e.g., to Sobolev classes, a common assumption class in non-parametric statistics.

## 1.2. Organisation

We first, in Section 2, briefly introduce the optimal transport problem, and our bandit setting. In Section 3, we present the optimistic algorithm for learning the Kantorovich problem, introducing the frequency-domain representation, the confidence sets, and the regularised optimism which define the algorithm. In Section 4, we study trajectorial regret bounds for the algorithm and discuss its generalisation to other bilinear problems. Finally, in Section 5 we discuss functional regression, including relevant regularity conditions for our setting and derive our worst-case regret bounds. Each section is expanded in a corresponding appendix.

## 2. Preliminaries

In this section, we will present the optimal transport problem and introduce our bandit learning setting. As we go, we will outline relevant elements of the literature and pinpoint the key challenges, laying the ground work for our algorithmic design and regret-analysis contributions in the following sections.[2]

### 2.1. The optimal transport problem

Optimal Transport (OT) is a mathematical theory that originated in the 18th century to optimise logistics (Monge, 1781; Kantorovich, 2006). Over the last few decades, however, this theory has experienced a meteoric rise in applied mathematics due to a sustained series of major breakthroughs (Brenier, 1989; Villani, 2003). On the back of this, it has also seen great success in applications from economics (e.g. Galichon, 2021; Kreinovich et al., 2024), generative modelling (Arjovsky et al., 2017), domain adaptation (Courty et al., 2017), and learning theory (see the survey of Chewi et al., 2024). In this section, in the interest of time, we present only the objective which will be studied in this paper, and refer the interested readers to the standards monographs (Ambrosio et al., 2021; Villani, 2009) for a broader overview of the field of optimal transport.

Given a pair or probability measures $(\mu, \nu) \in \mathscr{P}(\mathcal{M}_\mu) \times \mathscr{P}(\mathcal{M}_\nu)$, on two topological measurable spaces $(\mathcal{M}_\mu, \mathcal{F}_\mu)$ and $(\mathcal{M}_\nu, \mathcal{F}_\nu)$, the Kantorovich OT problem aims to optimise the cost of transporting $\mu$ to $\nu$ with respect to a cost function $c : \mathcal{M}_\mu \times \mathcal{M}_\nu \to \mathbb{R}$. For ease of exposition, we consider $\mathcal{X} := \mathcal{M}_\mu \times \mathcal{M}_\nu$ to be a subset of $\mathbb{R}^d$, $d \in \mathbb{N}$, but the problem below is also defined on highly esoteric spaces

---

[2]For complements and notations, see Appendices A and H.

$(\mathcal{M}_\mu, \mathcal{M}_\nu)$ such as a graph or a space of curves. Formally, the Kantorovich problem is defined as

$$\text{Kant.}(\mu, \nu, c) := \inf_{\pi \in \Pi(\mu, \nu)} \int c(x, y) \mathrm{d}\pi(x, y)$$

in which $\Pi(\mu, \nu)$, defined as

$$\{\pi \in \mathscr{P}(\mathcal{M}_\mu \times \mathcal{M}_\nu) : \pi(\cdot, \mathcal{M}_\nu) = \mu, \pi(\mathcal{M}_\mu, \cdot) = \nu\},$$

is the set of *transport plans* (or *couplings*) between $\mu$ and $\nu$. In other words, $\Pi(\mu, \nu)$ corresponds to all joint distributions whose first and second marginals are $\mu$ and $\nu$, respectively. Importantly, the Kantorovich problem allows for mass to be split and transported from a single point $x \in \mathcal{M}_\mu$ to multiple points $y \in \mathcal{M}_\nu$, but each set $S \in \mathcal{F}_\mu$ must give exactly $\mu(S)$ mass and each set $S' \in \mathcal{F}_\nu$ must receive exactly $\nu(S')$ mass.

Divisibility of mass was absent of the original formulation of the OT problem (Monge, 1781), which leads to a technically more challenging problem. In contrast, the Kantorovich problem is a linear program on the space of finite Radon measures $\mathscr{M}(\mathcal{X})$, in the sense that $\pi \in \mathscr{M}(\mathcal{X}) \to \int c(x, y) \mathrm{d}\pi(x, y)$ is a linear map and $\Pi(\mu, \nu)$ is defined by linear (integral) constraints. The Kantorovich problem is solvable when $c$ is lower semi-continuous and bounded below, see (Villani, 2009, Thm. 4.1), and, in fact, $\Pi(\mu, \nu)$ is convex and compact (Ambrosio et al., 2021, Cor. 2.9) despite $\mathscr{P}(\mathcal{X})$ not being a vector space.

If we consider the family of possible Kantorovich problems indexed by continuous cost functions $c$, and writing $\int c(x, y) \mathrm{d}\pi(x, y)$ as a duality bracket[3], we have

$$\text{Kant.}(\mu, \nu, c) = \inf_{\pi \in \Pi(\mu, \nu)} \langle c | \pi \rangle. \tag{3}$$

This is a bilinear form reminiscent of (2), except for the fact that the bracket $\langle \cdot | \cdot \rangle$ is not an inner product. Due to this (because $\pi$ and $c$ do not belong to a common Hilbert space) there are neither Riesz representation theorem nor Cauchy-Schwarz inequality that apply. Taking $J^\varsigma := \langle \varsigma | \cdot \rangle$, the Kantorovich problem lies precisely in the blindspot of the literature on linear bandits outlined in Section 1.

### 2.2. The bandit optimal transport (BOT) problem

Adapting the Kantorovich problem to the formulation of a stochastic bandit, we consider the following learning game. The agent knows $(\mathcal{M}_\mu, \mathcal{M}_\nu, \mu, \nu)$ ahead of time, so that it knows its action set, but it does not know the true cost function $c^*$. At each time step $t \in \mathbb{N}$, the agent must choose an admissible transport plan $\pi_t \in \Pi(\mu, \nu)$, and receives a noisy feedback $C_t = \int c(x, y) \mathrm{d}\pi_t(x, y) + \xi_t$, in which $(\xi_t)_{t \in \mathbb{N}}$ is a conditional martingale difference sequence.

---

[3]See Appendix B for details.

We evaluate its performance by the regret defined in (1), w.r.t. the Kantorovich problem Kant.$(\mu, \nu, c^*)$, i.e.

$$\mathscr{R}_T(\boldsymbol{\pi}) := \sum_{t=1}^{T} C_t - \text{Kant.}(\mu, \nu, c^*)$$

Note that this problem differs from *statistical* optimal transport and *online* optimal transport. We defer discussion of these related problems to Appendix H for brevity.

The first authors to take direct interest in the problem of *online learning* of OT appear to be Guo et al. (2022), who considered an online convex optimisation setting. In this setting, the cost function $c^*$ may change at each $t \in \mathbb{N}$ (while remaining suitably convex) but is revealed after each round. This work was followed by Zhu & Ryzhov (2023), who considered BOT only in a semi-discrete setting. They constructed a semi-myopic algorithm with forced exploration which can learn to behave as the optimal plan from samples of the (unknown) continuous marginal. Unfortunately, they study only the case in which $c^*$ is a parametric linear model and do not provide regret bounds. In contrast, we will leverage the intrinsic regularity of the Kantorovich problem to construct an optimistic algorithm which achieves sublinear regret bounds under minimal assumptions, including infinite-dimensional non-parametric problems.

## 3. Algorithmic Design

Let us now introduce our algorithm for learning the BOT problem. The method used to construct our algorithm is quite novel, despite being based on the celebrated OFU principle. Its three principle ingredients are:

**1)** Representing a subset of the action space in the same space as the hypothesis class of cost functions. We do this through the Fourier transform of both the measure and cost function, which turns (3) into an inner product on $\mathcal{H} := L^2(\mathcal{X}; \mathbb{C}; \varrho)$, the Hilbert space of square-integrable complex-valued functions on $\mathcal{X}$. Note that this transformation is valid *only* for measures with square-integrable densities, a dense subspace of $\mathscr{P}(\mathcal{M}_\mu \times \mathcal{M}_\nu)$. We develop this in Section 3.1.

**2)** In the resulting frequency domain, if all past actions have square-integrable densities, we can construct a Regularised Least Squares (RLS) estimator of the cost function $c^*$ at each time step $t \in \mathbb{N}$. This RLS estimator is then used to construct a confidence set $\mathcal{C}_t(\delta) \subset \mathcal{H}$ *à la* Abbasi-Yadkori (2012). This construction is presented in Section 3.2.

**3)** To preserve the coherence of this representation, we regularise the optimism step to optimise a penalised objective which acts as a barrier function which blows up as the square-integrability of the action's density vanishes. This regularisation is necessary to ensure that the algorithm only

takes actions with square integrable densities and thus that the future confidence sets $\mathcal{C}_s$ for $s > t$ remain valid. This last step is the subject of Section 3.3.

### 3.1. Frequency domain representation of actions

The technical issue with Kant.$(\mu, \nu, \cdot)$ comes from $\langle \cdot | \cdot \rangle$, which is not an inner product but the duality pairing between continuous functions vanishing at infinity and finite Radon measures, two different classes of objects. To reconcile these two types of objects, we will leverage Fourier analysis and represent them both in the frequency domain. To ensure the Fourier transform and the Kantorovich problem are well defined, let us assume Assumption 3.1.

**Assumption 3.1.** The true cost function $c^*$ is continuous and belongs to $L^2(\mathbb{R}^d; \varrho)$.

Let $\mathsf{F}$ denote the Fourier transform operator that acts on $L^2(\mathbb{R}^d; \varrho)$ or on finite Radon measures, using the formulæ

$$\mathsf{F} : \phi \in L^2(\mathbb{R}^d; \varrho) \mapsto \int \phi(x) e^{-2\pi i \langle x | \cdot \rangle_2} \mathrm{d}\varrho(x) \text{ and}$$

$$\mathsf{F} : \gamma \in \mathscr{M}(\mathbb{R}^d) \mapsto \int e^{-2\pi i \langle x | \cdot \rangle_2} \mathrm{d}\gamma(x),$$

when these are well defined (see Appendix B for detailed constructions), wherein $\langle \cdot | \cdot \rangle_2$ is induced by $\|\cdot\|_2$ on $\mathbb{R}^d$. The operator $\mathsf{F}$ is an isometry on $L^2(\mathbb{R}^d; \varrho)$, and thus

$$\langle c^* | \pi \rangle = \langle \mathsf{F}c^*(-\cdot) | \mathsf{F}\pi \rangle_{L^2(\mathbb{C}^d; \varrho)} := \int \mathsf{F}c^*(-z) \overline{\mathsf{F}\pi}(z) \mathrm{d}\varrho(z) \tag{4}$$

when this is well defined.

The right-hand side of (4) is well defined precisely when $\mathsf{F}\pi$ is in $L^2(\mathbb{R}^d; \varrho)$, which is equivalent to $\pi$ having a density $\mathrm{d}\pi/\mathrm{d}\varrho$ with respect to $\varrho$ which is itself in $L^2(\mathbb{R}^d; \varrho)$ (see Lemma B.4). This cannot be assumed *a priori* for any (let alone all) $\pi \in \Pi(\mu, \nu)$, but when (4) does hold, we can leverage it to construct confidence sets. We will return to the question of when this holds shortly, in Section 3.3.

Let $\Pi_{\mathscr{H}}(\mu, \nu) := \{\pi \in \Pi(\mu, \nu) : \mathrm{d}\pi/\mathrm{d}\varrho \in L^2(\mathbb{R}^d; \varrho)\}$ denote the subset of couplings which have a square-integrable density with respect to $\varrho$. We can embed this set into the Hilbert space $\mathscr{F} := L^2(\mathbb{R}^d; \varrho)$ by defining the embedding $\pi \hookrightarrow \mathsf{F}\mathrm{d}\pi/\mathrm{d}\varrho$. In $\mathscr{F}$, it is possible to define regularised least squares estimators and confidence sets using the classical theory, which we do in the next section.

### 3.2. Infinite-dimensional estimation of the cost function

The parameter space ($\mathscr{F}$) is unchanged under the Fourier transform, as $\mathsf{F}$ is an isometry, so that $\mathsf{F}c^* \in \mathscr{F}$, and likewise for $\mathsf{F}\Pi_{\mathscr{H}}(\mu, \nu)$. Consequently, up to reflection about $0$, we can express (4) as the inner product on the Hilbert space

$\mathscr{F}$. Thus we can use the concentration arguments of Abbasi-Yadkori (2012) to estimate $f^* := \mathsf{F}c^*(-\cdot)$. The validity of this methodology relies on the standard Assumption 3.2.

**Assumption 3.2.** An *a priori* scale estimate $\bar{C} \geq \|c^*\|_{L^2(\mathbb{R}^d;\varrho)}$ is known. The sequence $(\xi_t)_{t\in\mathbb{N}}$ is $\sigma^2$-sub-Gaussian for some $\sigma \in (0,+\infty)$.

Suppose a history of action $(\pi_t)_{s\leq t}$ and of observed costs $(C_t)_{s\leq t}$ is given at time $t$. To simplify notation, let us write $a_s := \mathsf{F}\pi_s$ for $s \in \mathbb{N}$, as well as $\boldsymbol{a}_t := (a_s)_{s\leq t}$ and $\boldsymbol{C}_t := (C_s)_{s\leq t}$, for $t \in \mathbb{N}$. Let us begin by defining the Regularised Least Squares (RLS) estimator of $f^*$. Let $\mathcal{L}.[\cdot] : \mathbb{N} \times \mathscr{F} \to \mathbb{R}$ be the (random) functional defined by

$$(t, f) \mapsto \mathcal{L}_t[f] := \sum_{s=1}^{t} \left\| C_s - \langle f|a_s\rangle_{L^2(\mathbb{R}^d;\varrho)} \right\|_2^2 .$$

Consider $\lambda > 0$ and a strongly convex and continuously Fréchet-differentiable regulariser $\Lambda : \mathscr{F} \to \mathbb{R}$ (e.g. $\frac{1}{2}\|\cdot\|_{L^2(\mathbb{R}^d;\varrho)}^2$), whose Fréchet derivative $\mathrm{D}\Lambda$ satisfies

$$\frac{1}{M_\Lambda}\|f\|_{L^2(\mathbb{R}^d;\varrho)} \leq \mathrm{D}\Lambda[f] \leq M_\Lambda\|f\|_{L^2(\mathbb{R}^d;\varrho)}$$

for any $f \in L^2(\mathbb{R}^d;\varrho)$, for some $M_\Lambda > 0$. Let us recall that the Fréchet derivative of a strongly convex Fréchet-differentiable functional is a (strongly) positive-definite operator denoted $\mathrm{D}\Lambda$.

It is clear that $\mathcal{L}_t + \lambda\Lambda$ is a strongly convex functional for any $\lambda > 0$ and $t \in \mathbb{N}$. Therefore, we can define the $\Lambda$-regularised least-squares (RLS) estimator of $f^*$ to be

$$\hat{f}_t^\lambda := \underset{f\in L^2(\mathbb{R}^d;\varrho)}{\operatorname{argmin}} \mathcal{L}_t[f] + \lambda\Lambda[f] . \qquad (5)$$

As in finite dimension, the RLS estimator can be obtained through a closed form expression, (see Proposition C.1). Let us define the family of linear a.s. bounded adjoint operators

$$M_t : f \in L^2(\mathbb{R}^d;\varrho) \mapsto (\langle f|a_t\rangle_{L^2(\mathbb{R}^d;\varrho)})_{i=1}^{t} \in \mathbb{R}^t ,$$

indexed by $t \in \mathbb{N}$, and their adjoints

$$M_t^* : v \in \mathbb{R}^t \mapsto \sum_{s=1}^{t} v_s a_s \in L^2(\mathbb{R}^d;\varrho) .$$

Let $V_t := M_t^* M_t$ and $V_t^\lambda := V_t + \lambda\mathrm{D}\Lambda$ denote the covariance and ridge-regularised covariance operators at time $t \in \mathbb{N}$. For every $\delta \in (0,1)$, define the confidence sets

$$\mathcal{C}_t(\delta) := \left\{ f \in \mathscr{F} : \left\| f - \hat{f}_t^\lambda \right\|_{V_t^\lambda} \leq \beta_t(\delta) \right\} , \qquad (6)$$

whose widths $\beta_t(\delta)$ are chosen as

$$\beta_t(\delta) := \sigma\sqrt{\log\left(\frac{4\det(\mathrm{D}\Lambda + \frac{1}{\lambda}M_t M_t^*)}{\delta^2}\right)} + \frac{\sqrt{\lambda}\bar{C}}{\|\mathrm{D}\Lambda\|_{\mathrm{op}}^{\frac{1}{2}}}. \qquad (7)$$

We show in Lemma C.2 that these sets are valid confidence sets for $f^*$ uniformly over time, with high probability. I.e. $f^* \in \cap_{t\in\mathbb{N}}\mathcal{C}_t(\delta)$ with probability at least $1 - \delta$. This completes the second ingredient of our algorithm.

### 3.3. Entropy-regularised optimism

In a standard application of optimism (e.g. OFUL in Abbasi-Yadkori, 2012), one would take actions

$$\pi_{t+1}^\circ \in \underset{\pi\in\Pi(\mu,\nu)}{\operatorname{argmin}} \min_{c\in\mathsf{F}^{-1}\mathcal{C}_t(\delta)} \langle c|\pi\rangle \qquad (8)$$

at each time $t \in \mathbb{N}$. This does not work with the confidence sets we have constructed for the optimal transport problem. Indeed, the minimiser of the right-hand side of (8) need not even have a density with respect to $\varrho$, let alone one in $L^2(\mathbb{R}^d;\varrho)$. Taking such an action would break the frequency-domain RLS fitting method and thus the guarantees of the confidence sets (i.e. Lemmas D.1 and C.2).

To circumvent this difficulty, we will intentionally limit our algorithm to plans which meet the density requirement, and thus accept sub-optimal solutions. To this end, we will penalise the transport cost by adding an entropic regularisation term, which we introduce next. Optimism with respect to this surrogate problem will incur an additional regret term, focused on later on in this section, when we will leverage results by Carlier et al. (2023) to show that it can be made a lower-order term.

Given a reference measure $\varrho \in \mathscr{P}(\mathbb{R}^d)$, the *entropy functional* (a.k.a. Kullback-Leibler divergence) is defined by

$$\pi \in \mathscr{P}(\mathbb{R}^d) \mapsto \mathscr{H}(\pi|\varrho) := \begin{cases} \int \log\frac{\mathrm{d}\pi}{\mathrm{d}\varrho}\mathrm{d}\pi & \text{if } \pi \ll \varrho \\ +\infty & \text{if } \pi \not\ll \varrho \end{cases} \quad (9)$$

where $\ll$ denotes absolute continuity. This functional is strictly convex and lower semi-continuous on $\mathscr{P}(\mathbb{R}^d)$, and finite if and only if $\pi$ has a density with respect to $\varrho$. The *Entropic Optimal Transport* (EOT) problem is then

$$\text{Ent.}(\mu,\nu,c,\varepsilon) := \inf_{\pi\in\Pi(\mu,\nu)} \langle c|\pi\rangle + \varepsilon\mathscr{H}(\pi|\varrho), \qquad (10)$$

where $\varepsilon > 0$ is a regularisation parameter. By virtue of $\mathscr{H}$, the functional $\Psi^\varepsilon : (c,\pi) \in L^2(\mathbb{R}^d;\varrho) \times \Pi(\mu,\nu) \to \langle c|\pi\rangle + \varepsilon\mathscr{H}(\pi|\varrho) \in \mathbb{R}$ is strictly convex and continuous, so that the infimum in (10) is attained. The EOT problem is a well-studied problem in optimal transport (see e.g. Peyré & Cuturi, 2019; Nutz, 2022).

The regularised optimism step of our algorithm will therefore take the form

$$\pi_{t+1} \in \underset{\pi\in\Pi(\mu,\nu)}{\operatorname{argmin}} \min_{c\in\mathsf{F}^{-1}\mathcal{C}_t(\delta)} \Psi^\varepsilon(c,\pi) \qquad (11)$$

for a sequence of regularisation parameters $(\varepsilon_t)_{t\in\mathbb{N}}$, going to $0$ with $t$, to be tuned in the regret (see below).

Regularising the optimism step by entropy, naturally, comes at a cost. There will be a regret term which accumulates the approximation errors over time, and thus we will need to change $\varepsilon$ over time to avoid constant regret. This regret term can be controlled by appealing to results, due to Carlier et al. (2023), about the convergence of the EOT problem to the Kantorovich problem as $\varepsilon \downarrow 0$. This result shows that

$$\mathrm{Ent.}(\mu,\nu,c^*,\varepsilon) - \mathrm{Kant.}(\mu,\nu,c^*) \leq C\varepsilon \log(\varepsilon^{-1})\,,$$

for some $C > 0$ depending on $\mu, \nu$ and the regularity of $c^*$. We detail this result and this constant in Appendix C.2.

### 3.4. Algorithm

Combining these ingredients into an algorithm, we arrive at Algorithm 1, which we dub `EntUCB` referencing its entropic penalisation and optimistic nature.

---

**Algorithm 1** `EntUCB`

---

**Require:** Confidence $\delta$, regularization level $\lambda$, entropy penalisation $(\varepsilon_t)_{t\in\mathbb{N}}$
1: Set $\tilde{\pi}_1 := \mu \otimes \nu$
2: **for** $t \in \mathbb{N}$ **do**
3:     Play $\pi_t = \tilde{\pi}_t$
4:     Receive feedback $C_t$
5:     Compute the RLS estimator $\hat{f}_t^\lambda$ using (5) or (18)
6:     Construct the confidence set $\mathcal{C}_t(\delta)$ using (6) and (7)
7:     **Optimism:** pick $(\tilde{\pi}_{t+1}, \tilde{c}_{t+1})$ according to (11)
8: **end for**

---

To illustrate the workings of Algorithm 1, we give some example applications in Appendix C.3. Let us now move on to the regret analysis in Section 4.

## 4. Trajectorial Regret Bounds

In this section, we give a trajectorial (i.e. dependent on $M_t$) regret analysis of Algorithm 1 for the BOT problem, which shows that `EntUCB` matches the bounds in classical linear bandits despite the infinite-dimensional and non-Hilbertian nature of the problem. We also show (deferred to Appendix D.2) that this result generalises to other linear functionals beyond the Kantorovich problem, under suitable regularity conditions.

### 4.1. Regret for bandit optimal transport

Let us turn first to the BOT problem and Algorithm 1. In Section 3.2, we showed that the confidence sets were valid in the frequency domain (see Lemma C.2). However, ensuring a confidence set is valid is not sufficient to ensure low regret, we also have to ensure that the size of this confidence set

is small enough to control the error of the agent. In the frequency domain, we can follow the standard arguments of Abbasi-Yadkori (2012) to bound the size of the confidence set in terms of the width $\beta_t(\delta)$, see Lemma D.1.

Let us now turn to the regret of Algorithm 1 with the proof of the trajectorial bound of Theorem 4.1. The first term is the trajectorial regret term standard in linear bandits, the second the cumulated approximation error due to entropic regularisation, and the last term the standard martingale concentration term.

**Theorem 4.1.** *Under Assumptions 3.1 and 3.2, if $c^*$ is $\alpha$-Hölder on $supp(\mu) \times supp(\nu) \subset \mathbb{R}^d$, for some $\alpha \in (0,1]$, then for any $\delta > 0$, $\lambda > 0$, $\eta \in (0,1)$, and $T \in \mathbb{N}$, the regret of Algorithm 1 with $(\varepsilon_t)_{t\in\mathbb{N}} = (\eta t^{-\eta})_{t\in\mathbb{N}}$, denoted $\mathcal{A}$, satisfies*

$$\mathscr{R}_T(\mathcal{A}) \leq 2\bar{C}\beta_T(\delta)\sqrt{T \log\det\left(\mathrm{Id} + \frac{M_T(\mathrm{D}\Lambda)^{-1}M_T^*}{2\lambda\bar{C}}\right)}$$
$$+ \frac{\kappa\eta}{1-\eta}\left(T^{1-\eta}\log(T) + \frac{\eta}{2^\eta}\log(6)\right)$$
$$+ \sigma\sqrt{2T\log\left(\frac{2}{\delta}\right)}$$

*with probability at least $1 - \delta$, wherein $\kappa$ is a constant dependant on $(\mu,\nu,\alpha)$ (see Lemma C.3).*

*Proof.* Core steps were described throughout Section 3, details are deferred to Appendix D. $\square$

If $\eta$ is chosen in $(1/2, 1)$, Theorem 4.1 the trajectorial term is dominant and the bound matches (Abbasi-Yadkori, 2012), showing that the BOT problem follows the same fundamental structure as the classical Hilbert space linear bandits. Notice that this result required significantly more work than Hilbert space linear bandits, due to the need to find an approximation regime in which a Hilbertian structure is valid. That this approximation can be made a lower order term in the regret is a consequence of inherent regularity of the BOT problem, as shown by Lemma C.3.

In Appendix D.2, we draw on the Kantorovich problem to derive a generalisation of this argument to other linear functionals, but the question of the generality of regularity properties akin to Lemma C.3 remains open. Next, in Section 5, we turn to the statistical question of estimation of the cost functional, which will allow us to show worst-case regret bounds for the BOT problem, and to show that the BOT problem is sub-linearly learnable even in infinite dimension.

# 5. Non-parametric Estimation and Worst-case Regret

We complete the BOT picture by deriving a learning protocol which gives precise control over the worst-case regret bounds in Theorem 5.3. Before giving these regret bounds, we introducing basis decomposition and the associated regularity condition, and integrate it into `EntUCB`.

## 5.1. Basis decomposition and regularity conditions

Since, in general, the frequency-domain representation $f^*$ of the cost function $c^* \in L^2(\mathbb{R}^d; \varrho)$ is an infinite-dimensional object, we must choose a basis in which to express and learn it. Classical choices of bases are wavelet systems, such as the Haar system, or the Fourier basis if $\text{supp}(\mu) \times \text{supp}(\nu)$ is bounded. Regardless of the choice made, for any orthonormal basis $(\phi_i)_{i \in \mathbb{N}}$ of $L^2(\mathbb{R}^d; \varrho)$, we can write

$$f^* = \mathsf{F}c^* = \sum_{i=1}^{+\infty} \gamma_i^* \phi_i \,,$$

for some sequence $\gamma^* \in \ell_2(\mathbb{R})$, with $\|\gamma^*\|_{\ell_2(\mathbb{R})} = \|f^*\|_{L^2(\mathbb{R}^d;\varrho)}$. The efficiency with which this basis represents $f^*$ will determine the convergence rate of the estimator for $f^*$ and thus the worst-case regret. To quantify this efficiency, we will take an approximation theory viewpoint with Assumption 5.1.

**Assumption 5.1.** There is a known orthonormal basis $(\phi_i)_{i \in \mathbb{N}}$ of $L^2(\mathbb{R}^d; \varrho)$, and $\zeta : \mathbb{R}_+ \to [0, 1]$, a known monotonically increasing continuous function satisfying

$$\inf_{n \in \mathbb{N}} \frac{\sum_{i=1}^n |\gamma_i^*|^2}{\zeta(n)} \geq \|c^*\|_{L^2(\mathbb{R}^d;\varrho)}^2 \,.$$

The formulation of Assumption 5.1 is inspired by the work of Kolmogorov in his study of approximations by finite sub-spaces (Kolmogorov, 1991). We recall this formulation, essential notions, and examples in Appendix E.1.

Henceforth, in the chosen orthonormal basis $(\phi_i)_{i \in \mathbb{N}}$ of $L^2(\mathbb{R}^d; \varrho)$, let each $a_t$ admit the representation

$$a_t = \sum_{i=0}^{\infty} \vartheta_i^{(t)} \phi_i \,, \quad \text{for some} \quad \vartheta^{(t)} \in \ell_2(\mathbb{R}) \,.$$

Since $(\phi_i)_{i \in \mathbb{N}}$ is an orthonormal basis, we may write

$$\langle f^* | a_t \rangle_{L^2(\mathbb{R}^d;\varrho)} = \langle \gamma^* | \vartheta^{(t)} \rangle_{\ell_2(\mathbb{R})} = \sum_{n=1}^{+\infty} \gamma_n \vartheta_n^{(t)} \,.$$

Let $f^*|_n := \sum_{i=0}^n \gamma_i^* \phi_i$ be the truncation of the basis expansion of $f$ at order $n \in \mathbb{N}$. By abuse of notation, and only when it is clear from context, we will override notation

and denote $c^*|_n$ the result of applying the inverse fourier transform to $f^*|_n$, the basis truncation of $f^* := \mathsf{F}c^*$. A straightforward derivation (see Appendix E.2) yields the approximation bound of Lemma 5.2.

**Lemma 5.2.** Let $(\phi_i)_{i \in \mathbb{N}}$ be an orthonormal basis of $L^2(\mathbb{R}^d; \varrho)$, and let $f \in L^2(\mathbb{R}^d; \varrho)$ with $f := \sum_{i=0}^{\infty} \gamma_i \phi_i$. Then, for every $g \in L^2(\mathbb{R}^d; \varrho)$, we have

$$\left| \langle f - f|_n | g \rangle_{L^2(\mathbb{R}^d;\varrho)} \right| \leq \|g\|_{L^2(\mathbb{R}^d;\varrho)} \sqrt{\sum_{i=n+1}^{+\infty} |\gamma_i|^2} \,.$$

For our purpose, $g = a_t$ is bounded by 1 in norm since $\|\mathsf{F}\pi_t\|_\infty \leq \pi_t(\mathbb{R}^d) = 1 = \varrho(\mathbb{R}^d)$, so that the resulting approximation error of $f^*$ by $(f^*|_n)_{n \in \mathbb{N}}$ is controlled entirely by the decay of the coefficients $(\gamma_i^*)_{i \in \mathbb{N}}$. Consequently, regret analysis can leverage Lemma 5.2 to move the problem to finite-dimensional regression on the coefficients $(\gamma_i^*)_{i \in \mathbb{N}}$.

The error bound of Lemma 5.2 is cumulated at each time step in the regret. Thus, to ensure sub-linear regret, it is necessary to increase the order $n_t$ of the RLS estimate $\hat{\gamma}_t^{n_t, \lambda}$ of $\gamma$ over time. We detail this estimator, construct its confidence sets $\tilde{C}_t^{n_t}(\delta)$, and prove technical lemmata in Appendix E. Integrating these elements into `EntUCB` yields Algorithm 2, whose regret bound is given in Theorem 5.3.

---

**Algorithm 2** `Basis-truncation EntUCB`

---

**Require:** Confidence $\delta$, regularisation level $\lambda$, entropy penalisation $(\varepsilon_t)_{t \in \mathbb{N}}$, approximation orders $(n_t)_{t \in \mathbb{N}}$
1: Set $\tilde{\pi}_1 := \mu \otimes \nu$
2: **for** $t \in \mathbb{N}$ **do**
3:      Play $\pi_t = \tilde{\pi}_t$
4:      Receive feedback $C_t$
5:      Compute the RLS estimator $\hat{\gamma}_t^{n_t, \lambda}$ using (31)
6:      Construct the confidence set $\tilde{C}_t^{n_t}(\delta)$ using (32) and (33)
7:      **Optimism:** pick $(\tilde{\pi}_{t+1}, \tilde{\gamma}_{t+1}^{n_t})$ according to (34)
8:      Receive feedback $R_t$
9: **end for**

---

**Theorem 5.3.** *Assume Assumptions 3.1 to 3.2 and $\zeta(n) = 1 - n^{-q}$ for some $q > 0$. For any $\delta \in (0, 1)$, $\lambda > 0$, $\varepsilon > 0$, let $\tilde{\mathcal{A}}$ denote Algorithm 2 with $(n_t)_{t \in \mathbb{N}} = (\lceil t^{\frac{1}{q+2}} \rceil)_{t \in \mathbb{N}}$, $\Lambda_n = \frac{1}{2} \|\cdot\|_2^2$, for all $n \in \mathbb{N}$, and $(\varepsilon_t)_{t \in \mathbb{N}} = (\eta t^{-\eta})_{t \in \mathbb{N}}$. For*

*any $T \in \mathbb{N}$, the following regret bound holds:*

$$
\mathscr{R}_T(\tilde{\mathcal{A}}) \leq \bar{C}\left(1 + \frac{2(q+2)}{q}T^{\frac{q+4}{2q+4}}\right) + \kappa(1 + \sqrt{T}\log(T))
$$

$$
+ 2\bar{C}\sigma T^{\frac{q+4}{2q+4}}\left(\sqrt{2\log\left(\frac{\lambda^{-1} + 2T^{\frac{q+1}{q+2}}\bar{C}^2}{\delta}\right)} + \sqrt{\lambda}\bar{C}\right)
$$

$$
\times \sqrt{\log\left(1 + \frac{2T^{\frac{q+1}{q+2}}}{\bar{C}^2}\right)} + \sigma\sqrt{2T\log\left(\frac{2}{\delta}\right)}.
$$

*Proof.* The proof builds on Theorem 4.1 augmented by technical lemmata on functional regression with basis truncation. The key step (Lemma E.2) is a diagonalisation argument (over $(n, t)$) on the confidence sets $\tilde{\mathcal{C}}_t^n(\delta)$ for both coverage and width. Details are deferred to Appendix E. $\square$

Theorem 5.3 shows that the regret of Algorithm 1 is controlled by the regularity of the cost function $c^*$. The regret bound can vary from $\mathcal{O}(\sqrt{T})$ for $q \to +\infty$ down to $\mathcal{O}(T)$ as $q \downarrow 0$. Note that $q = 0$ corresponds to $c^*$ being an indicator function, in which learning the anything is clearly impossible. The bound can be instantiated under different assumptions on $c^*$ (i.e. choices of $\zeta$ in Assumption 5.1). Corollary 5.4 gives one such example.

**Corollary 5.4.** *Under Assumptions 3.1 and 3.2, if $c^*$ satisfies*

$$
\int \left[(1 + \|z\|)^m c^*(z)\right]^2 \mathrm{d}\varrho(z) < +\infty
$$

*the regret bounds of Theorem 5.3 hold with $q = m$.*

### 5.2. Finite-dimensional (parametric) problems

We will now discuss some examples in which a finite basis of $N$ terms is sufficient to control the approximation error. Proposition 5.5 shows that the BOT problem indeed reaches the parametric rate ($\tilde{\mathcal{O}}(\sqrt{T})$) of finite dimensional linear bandits given only knowledge of upper bounds on $N$ and $\|c^*\|_{L^2(\mathbb{R}^d, \varrho)}$, and of the appropriate basis $(\phi_i)_{i \in \mathbb{N}}$.

**Proposition 5.5.** *Under Assumption 3.1, 3.2, and 5.1, with $\zeta = \mathbb{1}_{\{\cdot \geq N\}}$ for some $N \in \mathbb{N}$ (i.e. $\gamma_i^* = 0$ for all $i > N$), then under the conditions of Theorem 5.3 with $(n_t)_{t \in \mathbb{N}} \equiv N$, $\Lambda_n = \|\cdot\|_{L^2(\mathbb{R}^d;\varrho)}/2$, and $\eta = 1/2$ the bounds of Theorem 5.3 become*

$$
\mathscr{R}_T(\mathcal{A}) \leq 2\bar{C}N\sqrt{T}\log\left(\frac{1}{\lambda} + \frac{T\bar{C}^2}{N}\right) + \kappa(1 + \sqrt{T}\log(T))
$$

$$
+ \sigma\sqrt{2T\log\left(\frac{2}{\delta}\right)}
$$

*Proof.* By the same arguments as in Theorem 5.3. $\square$

In Appendix E.3, we further detail several examples of such finite-dimensional problems.

## 6. Conclusion

This paper studied the problem of linear bandits in non-Hilbert spaces (i.e. where the objective is bilinear but not an inner product) through the lens of an example: the Kantorovich optimal transport problem.

Despite the inability to construct confidence sets of bounded width through self-normalised inequalities in this setting, we leveraged a sub-space ($\Pi_{\mathscr{H}}(\mu, \nu)$) which admits a frequency-domain Fourier representation in $L^2(\mathbb{R}^d; \varrho)$ to construct confidence sets and control their width in the covariance-induced norm (Lemma C.2). In order to preserve the coherence of these confidence sets during learning, our algorithm regularised the optimism step of OFUL to ensure actions remain within $\Pi_{\mathscr{H}}(\mu, \nu)$ at all times. We then controlled the resulting approximation error using known quantitative estimates for the $\Gamma$-convergence of the entropic OT problem to the Kantorovich one (Lemma C.3).

We showed that the resulting algorithm, Algorithm 1, achieves the same trajectorial regret bounds as OFUL (Abbasi-Yadkori, 2012) despite the non-Hilbertian nature of the problem in Theorem 4.1. Furthermore, using functional regression techniques, we showed that the algorithm can be adapted (Algorithm 2) to achieve regret bounds interpolating between $\tilde{\mathcal{O}}(\sqrt{T})$ and $\tilde{\mathcal{O}}(T)$, depending on the regularity of the cost function (Theorem 5.3). The tight (up to logarithmic factors) rate $\tilde{\mathcal{O}}(p\sqrt{T})$ is achieved for parametric problems in dimension $p$ (Proposition 5.5), while a non-parametric assumption on the decay rate of the Fourier transform of the cost function in a relevant basis yields regret bounds of the form $\tilde{\mathcal{O}}(T^{\frac{q+4}{2q+4}})$ for $q > 0$ (Corollary 5.4) for the general non-parametric case.

Taken together, these results provide a first step towards a general theory of linear bandits in non-Hilbert spaces, and towards settling the question of the extent to which linearity is sufficient to achieve the rates of "linear" bandits or if an inner product structure is required. In Appendix D.2, we showed that a representation of a sub-space with sufficiently fast $\Gamma$-convergence of a barrier regularisation to the original problem is sufficient to achieve the same regret bounds as OFUL. However, it remains to be seen if *this* is a necessary condition, or if the results can be extended yet further to more general sub-spaces.

Indeed, the problem of $\Gamma$-convergent approximations raises the question of whether entropic regularisation is a universal tool for bilinear problems involving measures, or if it is specific to the Kantorovich problem. In terms of bandit theory, the resolution of the BOT problem also raises interesting questions, notably whether knowledge of $(\mu, \nu)$ can be fore-

gone, and of the extension to the Monge formulation of OT. We develop these question in more detail in Appendix G.

Like with all optimistic algorithms, the computational tractability of Algorithm 2 should be met with some scepticism. If $\text{supp}(\mu)$ and $\text{supp}(\nu)$ are finite, naturally, optimism is implementable. For continuous measures, the key question is the ability to discretise the problem while preserving the ability to control the regret of the approximation error. Promising directions exist in this regard using Voronoi partitions and the shadow of a coupling, see (**?**), but the problem of computationally efficient optimism remains open. In both cases, a point of contention is the optimisation algorithm used to solve the regularised optimism step, which must output a feasible plan in finite time, with a guarantee on the approximation error. This appears to be absent from the literature, as Sinkhorn's celebrated algorithm does *not* output a valid plan in finite time, only at the fixed point in the limit.

## Acknowledgements

The author would like to thank Nadav Merlis and Hugo Richard for their thoughtful comments on early versions of the manuscript, Austin J. Stromme for sharing his insights in statistical optimal transport, and Jaouad Mourtada for his references on Kolmogorov width. The author is most endebted to Hamish Flynn for rooting out a consequential typo in Lemma E.3.

## Impact Statement

This paper presents work whose goal is to advance the field of machine learning. There are many potential societal consequences of our work, none of which we feel must be specifically highlighted here.

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

# A. Organisation of Appendices

The following appendices are organised thematically and are mostly independent completions of various parts of the text or discussion of topics not mentioned therein for the sake of brevity. Appendix B provides a rigorous treatment of necessary Fourier analysis notions, which provides background for the details of Section 3.1. Appendices C to E contain the complements and detailed proofs of the main results of Sections 3 to 5, respectively. Appendix F collects some miscellaneous minor results used in the text. Appendix G contains more detailed discussions of the open problems mentioned in Section 6. Appendix H contains bibliographical notes on statistical optimal transport which readers unfamiliar with the field might find of interest to understand the context of the paper. It is a complement to Section 2.1.

## A.1. Notation

We gather here some standard notation which is used throughout the paper. For a finite positive Radon reference measure $\varrho$, let $L^p(\mathcal{X}, \mathbb{K}; \varrho)$, $p \in [1, \infty]$ and $\mathbb{K} \in \{\mathbb{R}, \mathbb{C}\}$, denote the space of functions $f : \mathcal{X} \to \mathbb{K}$ that are $p^{\text{th}}$-power integrable. When $\mathcal{X}$, $\mathbb{K}$, or $\varrho$ are clear from context we will drop them for brevity; by default $\mathbb{K} = \mathbb{C}$ is dropped. We allow complex functions ($\mathbb{K} = \mathbb{C}$) to deal with the Fourier transforms, but this has no noticeable effect as it does not impact the Hilbertian structure of the space $L^2(\mathbb{R}^d, \mathbb{K}; \varrho)$.

In the following, let $\langle \cdot | \cdot \rangle_{L^2(\mathbb{R}^d; \varrho)}$ denote the inner product on $L^2(\mathbb{R}^d; \varrho)$, $\langle \cdot | \cdot \rangle_{\ell_2(\mathbb{R}^d)}$ the one on $\ell^2(\mathbb{R}^d, \mathbb{K})$ (the space of square integrable $\mathbb{K}$-valued sequences) with $\|\cdot\|_{\ell_2(\mathbb{R}^d)}$ denoting its associated norm. On $\mathbb{R}^d$, $\langle \cdot | \cdot \rangle_2$ denotes the inner product, $\|\cdot\|_2$ the Euclidean norm. As before, let $\langle \cdot | \cdot \rangle$ denote the duality pairing between $\mathcal{M}(\mathbb{R}^d)$ (the space of finite Radon measures) and $\mathcal{C}_0(\mathbb{R}^d)$ (the space functions vanishing at infinity). The operator norm of a linear operator $A$ (in finite or infinite dimension) is denoted by $\|A\|_{\text{op}}$.

Throughout, all probabilistic statements are understood as holding in the filtered probability space $(\Omega, \mathcal{F}_\infty, \mathbb{F}, \mathbb{P})$, in which $\mathbb{F} := (\mathcal{F}_t)_{t \in \mathbb{N}}$ is the natural filtration of $(\xi_t)_{t \in \mathbb{N}}$, and $\mathcal{F}_\infty = \lim_{t \to \infty} \mathcal{F}_t$.

For two measures $(\gamma, \rho) \in \mathcal{M}(\mathbb{R}^d)$, $\gamma \ll \rho$ denotes that $\gamma$ is absolutely continuous with respect to $\rho$, in which case we use $\mathrm{d}\gamma/\mathrm{d}\rho$ to denote the Radon-Nikodym derivative (a.k.a. the density) of $\gamma$ with respect to $\rho$. Moreover, $\otimes$ denotes the tensor product, i.e. $\mu \otimes \nu$ is the product measure of $\mu$ and $\nu$.

The join and meet operators on $\mathbb{R}$ are denoted by $\vee$ and $\wedge$, respectively, so that for $x, y \in \mathbb{R}$, $x \vee y = \max(x, y)$ and $x \wedge y = \min(x, y)$.

# B. Elements of Fourier Analysis

We regroup in this appendix some standard results in Fourier analysis which will allow us to rigourously define the construction of Section 3.1. Note that we will eschew the standard notations $\hat{f}$ and $\hat{\gamma}$ in favour of $\mathsf{F}f$ and $\mathsf{F}\gamma$ to avoid confusion with the least-squares estimator, which we will denote using its standard hat.

## B.1. Formal definitions

To define the Fourier transform on $L^2(\mathbb{R}^d; \varrho)$, we will extend it from a dense subspace (see Definition B.1) of $L^2(\mathbb{R}^d; \varrho)$ to the whole space. This technical construction arises as a consequence of the fact that $L^2(\mathbb{R}^d; \varrho) \not\subset L^1(\mathbb{R}^d; \varrho)$ when $\varrho$ need not be finite[4], meaning the right-hand side of (12) may not be defined and $\mathsf{F}$ is ill-posed on $L^2(\mathbb{R}^d; \varrho)$, despite the fact that (12) is well-posed for $f \in L^1(\mathbb{R}^d; \varrho)$. The following is summarised from Constantin (2016, Ch.5–6), refer therein for a more detailed treatment or, e.g., to Folland (1992).

**Definition B.1.** The Schwartz space $\mathcal{S}(\mathbb{R}^d)$ is defined as

$$\left\{ \phi \in \mathcal{C}^\infty(\mathbb{R}^d; \mathbb{C}) : \sup_{x \in \mathbb{R}^d} |x^\alpha \partial_\beta \phi(x)| < +\infty \text{ for any } \alpha, \beta \in \mathbb{N}^d \right\}$$

in which $\alpha, \beta \in \mathbb{N}^d$ are multi-indices so that $x^\alpha := (x_i^{\alpha_i})_{i=1}^d$, and $\partial_\beta := \partial_{x_1}^{\beta_1} \cdots \partial_{x_d}^{\beta_d}$.

Note that $\mathcal{S}(\mathbb{R}^d)$ is a dense subspace of $L^2(\mathbb{R}^d; \varrho)$ and $L^1(\mathbb{R}^d; \varrho)$ as it contains $\mathcal{C}_c^\infty(\mathbb{R}^d; \mathbb{C})$ the space of infinitely-differentiable compactly-supported (a.k.a. test) functions, which is dense in both $L^2(\mathbb{R}^d; \varrho)$ and $L^1(\mathbb{R}^d; \varrho)$.

---

[4]Despite using finite measures in the main text, we provide the general case to avoid hiding difficulties.

**Theorem B.2** ((Constantin, 2016, Thm. 6.1)). *Consider the Fourier transform operator* $\mathsf{F}$ *on the Schwartz space, with*

$$\mathsf{F} : \phi \in \mathcal{S}(\mathbb{R}^d) \mapsto \int \phi(x) e^{-2\pi i \langle x | \cdot \rangle_2} \mathrm{d}\varrho(x) \,. \tag{12}$$

*This operator maps* $\mathcal{S}(\mathbb{R}^d)$ *maps onto itself and is an isometric bijection. Moreover,*

$$\mathsf{F}^{-1} = \mathsf{F}\mathsf{R} \,, \tag{13}$$

*in which* $\mathsf{R} : \phi \in \mathcal{S}(\mathbb{R}^d) \mapsto \phi(-\cdot) \in \mathcal{S}(\mathbb{R}^d)$ *is the* reflection *operator.*

**Theorem B.3** ((Constantin, 2016, Thm. 6.4)). *The fourier transform* $\mathsf{F}$ *can be extended to a unitary operator on* $L^2(\mathbb{R}^d; \varrho)$ *and* (13) *holds on* $L^2(\mathbb{R}^d; \varrho)$ *for this extension.*

The formal inversion property (13) is easily shown to recover the classical inversion formula

$$f(x) = \int \mathsf{F}f(\xi) e^{2\pi i \langle x | \xi \rangle} \mathrm{d}\varrho(\xi) \text{ for } \varrho\text{-a.e. } x \in \mathcal{X} \tag{14}$$

as soon as $f \in L^1(\mathbb{R}^d; \varrho) \cap L^2(\mathbb{R}^d; \varrho)$ (recall that, in our case, $\varrho$ is a finite measure so $L^2(\mathbb{R}^d; \varrho) \subset L^1(\mathbb{R}^d; \varrho)$ and the inversion formula always holds). If $\varrho$ is only $\sigma$-finite (e.g. the Lebesgue measure), one must take slightly higher care. Namely the difference between (13) and (14) is whether the integral in (14) is well defined for $f \in L^2(\mathbb{R}^d; \varrho)$, which is not guaranteed.

This technicality reflects the limits used in the definition of the extension which are hidden by the abstract statement of Theorem B.3. Nevertheless, since the Schwartz space $\mathcal{S}(\mathbb{R}^d)$ is dense in both $L^1(\mathbb{R}^d; \varrho)$ and $L^2(\mathbb{R}^d; \varrho)$, we can always take an arbitrarily close function in $\mathcal{S}(\mathbb{R}^d)$ and invert that, the result will remain arbitrarily close in $L^2(\mathbb{R}^d; \varrho)$.

The Schwartzian framework is a robust one for Fourier analysis more generally, which can be used to extend $\mathsf{F}$ beyond $L^2(\mathbb{R}^d; \varrho)$. In particular, it can be used to unify the definitions we gave for the Fourier transform of a function and a measure (refer to Constantin, 2016, § 6.1.2, for more details). Precisely, one extends to the topological dual of $\mathcal{S}(\mathbb{R}^d)$ (the space of tempered distributions $\mathcal{S}'(\mathbb{R}^d)$), which includes $\mathscr{M}(\mathbb{R}^d)$ and $L^2(\mathbb{R}^d; \varrho)$ as sub-spaces.

A fundamental consequence of the various formulations of the Fourier transform is that measures whose transforms are in $L^2(\mathbb{R}^d; \varrho)$ are exactly those which have an $L^2(\mathbb{R}^d; \varrho)$ density with respect to $\varrho$. We will continue to denote the density of a measure $\mu$ with respect to $\varrho$ using the Radon-Nikodym notation $\mathrm{d}\mu/\mathrm{d}\varrho$, even when this tempered distribution can be identified with a function.

**Lemma B.4.** *Let* $\gamma \in \mathscr{M}(\mathcal{X})$ *be a finite Radon measure, if it has density with respect to* $\varrho$ *and* $\mathrm{d}\gamma/\mathrm{d}\varrho \in L^2(\mathbb{R}^d; \varrho)$, *then*

$$\mathsf{F}\gamma = \mathsf{F}\frac{\mathrm{d}\gamma}{\mathrm{d}\rho} \in L^2(\mathbb{R}^d; \varrho) \,.$$

*Conversely, if* $\mathsf{F}\gamma \in L^2(\mathbb{R}^d; \varrho)$, *then* $\gamma$ *has a density with respect to* $\varrho$ *and* $\mathrm{d}\gamma/\mathrm{d}\varrho \in L^2(\mathbb{R}^d; \varrho)$.

*Proof.* The first part is a direct consequence of the definitions of the Fourier transforms of a measure and an $L^2(\mathbb{R}^d; \varrho)$ function. For the converse, the fact that $\mathsf{F}\gamma \in L^2(\mathbb{R}^d; \varrho)$ implies $\gamma \ll \varrho$ involves some technical minutiae due to the different topologies $\mathscr{M}(\mathcal{X})$ can be equipped with, which we won't reproduce for conciseness, refer, e.g., to Lemma 1.1 of Fournier & Printems (2010). That the density is then in $L^2(\varrho)$ is a simple consequence of Plancherel's theorem:

$$\left\| \frac{\mathrm{d}\gamma}{\mathrm{d}\rho} \right\|_{L^2(\mathbb{R}^d; \rho)} = \int_{\mathbb{R}^d} |F\gamma(\xi)|^2 \, \mathrm{d}\rho(\xi) = \|F\gamma\|_{L^2(\mathbb{R}^d; \rho)} \,.$$

$\square$

## B.2. Fourier transform of measures

Let $\mathcal{C}_0(\mathbb{R}^d, \mathbb{K})$ denote the space of continuous functions from $\mathbb{R}^d$ to $\mathbb{K} \in \{\mathbb{R}; \mathbb{C}\}$, $\mathscr{M}(\mathbb{R}^d)$ denote the space of finite Radon measures over $\mathbb{R}^d$, and let us define the Fourier operator on this space by using the same notation, i.e. $\mathsf{F} : \gamma \in \mathscr{M}(\mathbb{R}^d) \mapsto \mathsf{F}\gamma \in \mathcal{C}_0(\mathbb{R}^d; \mathbb{C})$ with

$$\mathsf{F}\gamma : \xi \in \mathbb{R}^d \mapsto \int e^{-2\pi i \langle x | \xi \rangle_2} \mathrm{d}\gamma(x) \,. \tag{15}$$

The Riesz-Markov theorem shows that $(\mathscr{M}^*(\mathbb{R}^d), \|\cdot\|_\infty)$, the space of finite signed Radon measures on $\mathbb{R}^d$ (endowed with the total variation norm $\|\cdot\|_\infty$), is the topological dual of $(\mathcal{C}_0(\mathbb{R}^d), \|\cdot\|_\infty)$, the space of continuous functions which vanish at infinity (endowed with the supremum norm $\|\cdot\|_\infty$), refer, e.g., to Constantin (2016, p. 242). This duality is characterised by the pairing

$$\langle \cdot | \cdot \rangle : (f, \gamma) \in \mathcal{C}_0(\mathbb{R}^d) \times \mathscr{M}^*(\mathbb{R}^d) \mapsto \int f \mathrm{d}\gamma \in \mathbb{R}.$$

This pairing applies in particular to all functions $f \in \mathcal{C}(\mathcal{X}; \mathbb{R})$ if $\mathcal{X}$ is compact and to all positive finite Radon measures $\gamma \in \mathscr{M}^+(\mathcal{X})$, and we will use the pairing notation in this case too. In general we will use the notation for arbitrary functions, understood that it will be well defined (see also Remark B.6). In particular:

$$\text{Kant.}(\mu, \nu, c) = \inf_{\pi \in \Pi(\mu, \nu)} \langle c | \pi \rangle.$$

**Lemma B.5.** *For any finite positive Radon measures $\rho \in \mathscr{M}^+(\mathbb{R}^d)$ and $\gamma \in \mathscr{M}^+(\mathbb{R}^d)$ with $\mathrm{d}\gamma/\mathrm{d}\rho \in L^2(\mathbb{R}^d; \rho)$, and any $f \in L^2(\mathbb{R}^d; \rho) \cap L^1(\mathbb{R}^d; \rho)$, we have*

$$\langle f | \gamma \rangle = \langle \mathsf{FR}f | \mathsf{F}\gamma \rangle_{L^2(\mathbb{R}^d; \rho)}$$

*and*

$$|\langle f | \gamma \rangle| \le \|f\|_{L^2(\mathbb{R}^d; \rho)} \, \rho(\mathbb{R}^d) \gamma(\mathbb{R}^d)^2.$$

*Proof.* By (14),

$$\langle f | \gamma \rangle := \int f \mathrm{d}\gamma = \int \int \mathsf{F}f(\xi) e^{2\pi i \langle x | \xi \rangle} \mathrm{d}\rho(\xi) \mathrm{d}\gamma(x). \tag{16}$$

Let $\varphi : (x, \xi) \mapsto e^{2\pi i \langle x | \xi \rangle}$. Using (16), since by the Cauchy-Schwartz inequality

$$\begin{aligned}
|\langle f | \gamma \rangle| &\le \|Ff\|_{L^2(\mathbb{R}^d \times \mathbb{R}^d; \gamma \otimes \rho)} \|1\|_{L^2(\mathbb{R}^d \times \mathbb{R}^d; \gamma \otimes \rho)} \\
&= \|Ff\|_{L^2(\mathbb{R}^d; \rho)} \gamma(\mathbb{R}^d)^2 \rho(\mathbb{R}^d) < +\infty,
\end{aligned} \tag{17}$$

the integrand in (16) is $\gamma \otimes \rho$-integrable, and thus we can apply the Fubini-Lebesgue theorem to obtain

$$\langle f | \gamma \rangle = \int \mathsf{F}f(\xi) e^{2\pi i \langle x | \xi \rangle} \mathrm{d}[\gamma \otimes \rho](\xi, x) = \langle Ff | \varphi \rangle_{L^2(\mathbb{R}^d \times \mathbb{R}^d; \gamma \otimes \rho)}.$$

While, at the same time,

$$\begin{aligned}
\langle f | \gamma \rangle &= \int \mathsf{F}f(\xi) \int e^{2\pi i \langle x | \xi \rangle} \mathrm{d}\gamma(x) \mathrm{d}\rho(\xi) \\
&= \langle \mathsf{FR}f | \mathsf{F}\gamma \rangle_{L^2(\mathbb{R}^d; \rho)}.
\end{aligned}$$

By (17), we have once more:

$$|\langle f | \gamma \rangle| = \left| \langle \mathsf{FR}f | \mathsf{F}\gamma \rangle_{L^2(\mathbb{R}^d; \rho)} \right| \le \|Ff\|_{L^2(\mathbb{R}^d; \rho)} \gamma(\mathbb{R}^d)^2 \rho(\mathbb{R}^d).$$

$\square$

The benefit of Lemma B.5 may not be immediately apparent, but it is revealed when one notices that the $L^2(\mathbb{R}^d; \rho)$ inner products and norms considered on the right hand side depend only on the measure $\rho$ and not on $\gamma$. Thus, we are able to assume only integrability of $c^*$ only with respect to our reference measure $\varrho$ (recall (9)) and still manipulate the duality product $\langle c^* | \gamma \rangle$ for any $\gamma$. In particular, by taking $\varrho = \mu \otimes \nu$ given marginals $\mu$ and $\nu$ and playing $\pi_t$ such that $\Psi^\varepsilon_{\mu, \nu}(c^*, \pi_t) < +\infty$ (recall (10)) we can reduce $\langle c^* | \pi_t \rangle$ to a $L^2(\mathbb{R}^d; \varrho)$ inner product, moving our problem to a Hilbert space.

*Remark* B.6. Lemma B.5 opens the subject of discussing Assumption 3.1. Let us remark that if $S := \text{supp}(\mu \otimes \nu)$ is compact, continuity of $c^*$ on the closure of $S$ is sufficient to obtain these results. Similarly, if $c^*$ is bounded. However, Assumption 3.1 allows for many more functions, for instance it allows $c^* : (x, y) = \|x - y\|_2^2$ if $(\mu, \nu) \in \mathscr{P}_2(\mathbb{R}^d)$, where $\mathscr{P}_2(\mathbb{R}^d)$ denotes measures with a finite second moment. This is of value as it covers the Wasserstein distances which are of broad interest. In general, one can develop finer assumptions based on $(\mu, \nu)$ even if $\varrho$ is not finite, but we do not detail this for brevity.

## C. Complements to Algorithmic Design

In this section, we provide complements to the algorithmic design of Section 3, notably the construction of the least-squares estimator in the frequency domain and the validity of the confidence sets used in Algorithm 1.

### C.1. Confidence sets in $\Pi_{\mathscr{H}}$

We begin this section by giving a closed form for the regularised least-squares estimator $\hat{f}_t^\lambda$, which extends the standard finite-dimensional formula to infinite dimension in Proposition C.1. We then construct confidence sets around this estimator, and prove their uniform coverage in Lemma C.2.

**Proposition C.1.** *Assume Assumption 3.1, then for any $\lambda > 0$, and $t \in \mathbb{N}^*$, we have*

$$\hat{f}_t^\lambda = (M_t^* M_t + \lambda \mathrm{D}\Lambda)^{-1} M_t^* \boldsymbol{C}_t \,. \tag{18}$$

*Proof.* This proof extends the standard arguments for finite-dimensional least-squares, we include it for completeness focusing on the differences owing to infinite dimensions, cf. e.g. Abbasi-Yadkori (2012, § 3.2). One first computes the Fréchet derivative of $\mathcal{L}_t$, by studying a variation $\delta f \in L^2(\mathbb{R}^d; \varrho)$ and

$$\mathcal{L}_t[f + \delta f] - \mathcal{L}_t[f] \,.$$

One sees that the Fréchet derivative of $\mathcal{L}_t$ exists for all $t$ and is given by

$$f \mapsto \sum_{s=1}^t \left( \langle f | a_s \rangle_{L^2(\mathbb{R}^d; \varrho)} - C_s \right) a_s + \lambda \mathrm{D}\Lambda f = (M_t^* M_t + \lambda \mathrm{D}\Lambda) f - M_t^* \boldsymbol{C}_t \,.$$

Note that the right-hand side is easily checked by expanding the definition of $M_t$ and $M_t^*$, and in doing so one easily checks that $M_t^*$ is indeed the adjoint of $M_t$. Carrying on, by first order optimality, the normal equation is

$$(M_t^* M_t + \lambda \mathrm{D}\Lambda) \hat{f}_\lambda = M_t^* \boldsymbol{C}_t \,.$$

Since $M_t^* M_t$ is positive semi-definite and $\mathrm{D}\Lambda$ is positive definite, (18) follows. $\qquad \square$

To turn these confidence sets into a *bona fide* optimistic algorithm, we have to guarantee their validity uniformly in $t \in \mathbb{N}$. To this end, define the event

$$\mathcal{E}_t(\delta) := \left\{ \left\| \hat{f}_t^\lambda - \mathsf{F}c^* \right\|_{V_t^\lambda} \le \sigma \sqrt{\log\left( \frac{4 \det(\mathrm{D}\Lambda + \lambda^{-1} M_t M_t^*)}{\delta^2} \right)} + \left( \frac{\lambda}{\|\mathrm{D}\Lambda\|_{\mathrm{op}}} \right)^{\frac{1}{2}} \|\mathsf{F}c^*\|_{L^2(\mathbb{R}^d; \varrho)} \right\} \,. \tag{19}$$

for $t \in \mathbb{N}$. We will rely on the existing arguments of Abbasi-Yadkori (2012, Cor. 3.6), which yields Lemma C.2 below. While the confidence set of Lemma C.2 lies in the frequency domain, this doesn't pose any difficulty in the proof since $\mathsf{F}$ is a isometry on $L^2(\mathbb{R}^d; \varrho)$, so we can treat the position and frequency domains interchangeably from an analytical standpoint.

**Lemma C.2** ((Abbasi-Yadkori, 2012)). *For every $\delta \in (0, 1)$, $\lambda > 0$, under Assumptions 3.1 and 3.2 we have*

$$\mathbb{P}\left( c^* \in \bigcap_{t \in \mathbb{N}} \mathsf{F}^{-1} \mathcal{C}_t(\delta) \right) \ge \mathbb{P}\left( \bigcap_{t \in \mathbb{N}} \mathcal{E}_t(\delta/2) \right) \ge 1 - \frac{\delta}{2} \,.$$

*Proof.* Recall that $\mathsf{F}$ is an isometry on $L^2(\mathbb{R}^d; \varrho)$, and so is $\mathsf{F}^{-1}$, so $\mathsf{F}^{-1} \mathcal{C}_t$ is a confidence set for $c^*$ in $L^2(\mathbb{R}^d; \varrho)$, and it is an ellipsoid of identical radii centred at $\mathsf{F}^{-1} \hat{f}_t^\lambda$. A direct combination of Assumption 3.2, (19), and Cor. 3.6 of Abbasi-Yadkori (2012) yields

$$\mathbb{P}\left( \bigcap_{t \in \mathbb{N}} \mathcal{E}_t(\delta/2) \right) \ge 1 - \frac{\delta}{2} \,.$$

The second results follow by comparison of (19) and (7). $\qquad \square$

## C.2. Entropy-regularised optimism

This section is dedicated to the entropic regularisation term used in Algorithm 1, namely proving that the solutions to the entropic optimal transport (EOT) problem lie in $\Pi_{\mathscr{H}}$, so that (11) is well defined, and, thereafter, the approximation result of Lemma C.3.

We can easily show that the solutions to the EOT problem, and thus actions chosen with (11), yield measures with bounded densities with respect to $\varrho$, through the dual formulation of the EOT problem,

$$
\text{Ent.}(\mu, \nu, c, \varepsilon) = \sup_{(\varphi, \psi) \in \Xi} \left\{ \int \varphi \mathrm{d}\mu + \int \psi \mathrm{d}\nu - \varepsilon \int e^{\varepsilon^{-1}(\varphi + \psi - c)} \mathrm{d}(\varrho) + \varepsilon \right\}, \tag{20}
$$

wherein $\Xi := \{(\varphi, \psi) \in L^1(\mathbb{R}^d; \mu) \times L^1(\mathbb{R}^d; \mu) : \varphi \oplus \psi \le c\}$ with $\varphi \oplus \psi : (x, y) \mapsto \varphi(x) + \psi(y)$. From a dual solution $(\varphi^*, \psi^*) \in \Xi$, one may recover (see, e.g., Nutz, 2022, Thm. 4.2) a primal solution $\pi^*$ with density

$$
\frac{\mathrm{d}\pi^*}{\mathrm{d}\varrho} = e^{\frac{\varphi^* \oplus \psi^* - c}{\varepsilon}} < +\infty.
$$

Since $\varrho$ is a finite measure, $L^\infty(\mathbb{R}^d; \varrho) \subset L^2(\mathbb{R}^d; \varrho)$, so that $\mathsf{F}\pi^* \in L^2(\mathbb{R}^d; \varrho)$.

We will complete this section by recalling existing regularity results about the convergence of the EOT problem as $\varepsilon \to 0$, which will allow us to control this regret term in Section 4.

These results, due to Carlier et al. (2023), depend on the marginals through a complexity metric known as the *upper Renyi dimension*, which we denote by $d_{\mathscr{H}}(\gamma)$, for $\gamma \in \{\mu, \nu\}$, and which is defined by

$$
d_{\mathscr{H}}(\gamma) := \limsup_{\epsilon \downarrow 0} \frac{H_\varepsilon(\gamma)}{\log(\varepsilon^{-1})}
$$

in which $H_\varepsilon(\gamma)$ is the infimum (over all countable partitions of $\text{supp}(\gamma)$ into Borel subsets of diameter at most $\varepsilon$) of the discrete entropy of $\gamma$ with respect to the partition, see (Carlier et al., 2023). Using this quantity, Carlier et al. (2023) establish Lemma C.3 as a bound on the convergence of the EOT problem to the Kantorovich problem as $\varepsilon \downarrow 0$.

**Lemma C.3** (Carlier et al. (2023, Prop. 3.1, Rem. 3.4)). *If $c^*$ is $\alpha$-Hölder continuous on $supp(\mu) \times supp(\nu)$, then*

$$
\textit{Ent.}(\mu, \nu, c^*, \varepsilon) - \textit{Kant.}(\mu, \nu, c^*) \le \left( \frac{d_{\mathscr{H}}(\mu) \wedge d_{\mathscr{H}}(\nu)}{\alpha} + o(1) \right) \varepsilon \log(\varepsilon^{-1})
$$

*as $\varepsilon \downarrow 0$, in which $o(1)$ denotes a term which vanishes as $\varepsilon \to 0$.*

Extensions of this result exist to absolute continuity[5] conditions (see Carlier et al., 2023, Rem. 3.4). The rate in $\varepsilon$ is sharp in general (see Carlier et al., 2023, Ex. 3.6), but tighter bounds may be obtained under stronger regularity assumptions (see, e.g., Carlier et al., 2023, Prop. 3.7). In view of Lemma C.3, we will use the shorthand $\kappa := \alpha^{-1}(d_{\mathscr{H}}(\mu) \wedge d_{\mathscr{H}}(\nu)) + o(1)$. In spite of its apparent complexity, upper Renyi dimension is a relatively well behaved object, and can be bounded in many common situations, as seen in the following remarks.

*Remark* C.4 (Carlier et al. (2023, Prop. 3.2)). If $\gamma$ is a measure on $\mathbb{R}^d$ satisfying

$$
\int 0 \vee \log(\|x\|_2) \mathrm{d}\gamma(x) < +\infty
$$

then $d_{\mathscr{H}}(\gamma) \le d$.

*Remark* C.5 (Carlier et al. (2023, Rem. 3.5)). If $\gamma$ is finitely supported, then $d_{\mathscr{H}}(\gamma) = 0$.

## C.3. Example applications

To illustrate the workings of Algorithm 1 and close out this appendix, this section focuses on some examples of applications for `EntUCB`. We will focus exclusively on non-atomic measures, as discrete problems can be represented using classical finite-dimensional linear bandit techniques.

---

[5]For the purpose of our regret analysis, it is sufficient that the modulus of continuity of $c^*$ grows at a (strictly) super-logarithmic rate at infinity. While such functions technically form a greater class than Hölder functions, we will continue to refer to the Hölder condition for ease of exposition.

*Example* C.6 (Logistic optimisation). Consider the perspective of a delivery contractor which has just accepted a delivery contract in a new country. It must deliver goods on a regular basis from warehouses located around the country to various customers. In practice, these warehouses and customers are discrete, but we will represent them as generic measures, as a macroscopic view of the problem. Thus, let $\mu \in \mathscr{P}(\mathcal{X})$ and $\nu \in \mathscr{P}(\mathcal{X})$ be two measures on the space $\mathcal{X}$, representing the country, which is assumed to be a compact and connected Riemannian Manifold.

For the sake of argument, we assume the delivery contractor has no *a priori* knowledge of the costs of delivery, denoted by $c^* : \mathcal{X} \times \mathcal{X} \to \mathbb{R}$, which may depend on a variety of factors such as distance, road quality, traffic. Each day, a delivery plan $\pi_t \in \Pi(\mu, \nu)$ is implemented, which may be deterministic or random[6]

Algorithm 1 suggests beginning by distributing the goods at random, and then using the confidence sets to establish a set of credible costs. This behaviour is in line with the classical OFU principle. However, the entropic regularisation of optimism deviates from this intuitive template by forbidding the algorithm from ever using a deterministic plan, thus enforcing a degree of randomness in the delivery plans. This randomness ensures that the spectrum of the covariance operator of the actions does not blow up, which would make the confidence set diameter unbounded.

Despite this regularisation of the covariance operator, the choice of actions remains optimistic, which ensures that Algorithm 1 still addresses the exploration-exploitation trade-off inherent in this bandit problem.

From a modelling standpoint, there are many possible cost-functions one could consider realistic. Let us discussion the conditions of Lemma C.3 in regards to two of them. An intuitive formulation of the transport cost is to express it as the shortest path with respect to an underlying cost function $\ell : \mathcal{X} \to \mathbb{R}$, i.e.

$$c^*(x, x') = \inf_{\gamma \in \Gamma(x, x')} \int_0^1 \ell(\gamma(t))\dot{\gamma}(t)\mathrm{d}t$$

wherein $\Gamma(x, x')$ is the set of all smooth monotonous continuous curves $\gamma : [0, 1] \to \mathcal{X}$ such that $\gamma(0) = x$ and $\gamma(1) = x'$. If the cost $\ell$ is bounded on $\mathcal{X}$ and smooth, then the function $c^*$ will be Lipschitz continuous, satisfying the conditions of Lemma C.3 with $\alpha = 1$. This formulation of the cost function is also related to the dynamic version of the OT problem and to control theory, see, for instance, Benamou & Brenier (2000).

*Example* C.7 (Pricing of complex options). Let us consider an example owed to Galichon (2021, § 3.6) of a simple financial market consisting of two assets: $X$ and $Y$. The markets and options markets for $X$ and $Y$ are known to be complete and arbitrage-free with risk-neutral measures $\mu$ and $\nu$ respectively. Concretely, this means that the agent knows how to price any options on $X$ or $Y$ in such a way as to eliminate arbitrage opportunities, which one can consider a "fair" price.

However, this is not sufficient to price options which depend on the joint asset $(X, Y)$, for example an option to exchange $X$ for $Y$ at a future date. Pricing an option with payoff $u : [0, +\infty)^2 \to \mathbb{R}$, as Galichon highlights, amounts to solving

$$\max_{\pi \in \Pi(\mu, \nu)} \int u(x, y)\mathrm{d}\pi(x, y),$$

which is an optimal transport problem.

A bandit version of this problem can be encountered when the agent must price an option without knowing its exact payoff. While a financial institution is unlikely to accept the risk of offering a contract whose payoff is unknown, the statistical exercise is interesting. In this case, we can consider the institution offers the contract to the market each day, priced according to its guess for a risk-neutral measure $\pi_t$. After the contract has been sold and passed maturity, which we consider as one unit of time, the agent observes a realised payout $u(\hat{X}, \hat{Y})$ where $\hat{X}$ and $\hat{Y}$ are the realised values of the assets at maturity (copies of $X$ and $Y$, respectively). Assuming that the payoff $u(X, Y)$ has bounded variance uniformly under $\pi$ guarantees that this estimate is a sub-gaussian perturbation away from $\int u(x, y)\mathrm{d}\pi_t(x, y)$, so that the bandit feedback assumption is satisfied. Hölder continuity of the payoff function is a classic assumption in financial mathematics.

## D. Regret Bounds

This appendix is dedicated to the detailed proofs of the trajectorial regret bounds of Section 4, including technical lemmata. In Appendix D.1, we provide the proof of Theorem 4.1, while in Appendix D.2, we provide the generalisation to other linear functionals.

---

[6]A deterministic plan means all the mass from $x \in \mathcal{X}$ is transported to exactly one $x' \in \mathcal{X}$. In mathematical terms, there is a function $T : \mathcal{X} \to \mathcal{X}$ such that $\pi_t$ is supported on the graph of $T$.

### D.1. Trajectorial regret bounds for BOT

Before proving Theorem 4.1, we begin by proving Lemma D.1, which controls the main error term in the regret decomposition on the "good" event.

**Lemma D.1.** *Under Assumptions 3.1 and 3.2, on the event $\{c^* \in \cap_{t \in \mathbb{N}} \mathsf{F}^{-1}\mathcal{C}_t(\delta)\}$, for any $T \in \mathbb{N}$ and $(c_t)_{t=1}^T$ with $c_t \in \mathsf{F}^{-1}\mathcal{C}_t(\delta)$ for $t \in [T]$, we have*

$$\sum_{t=1}^T \langle c^* - c_t | \tilde{\pi}_t \rangle \leq 2\bar{C}\beta_T(\delta)\sqrt{T \log \det \left( \mathrm{Id} + \frac{1}{2\lambda\bar{C}} M_t (\mathrm{D}\Lambda)^{-1} M_t^* \right)}$$

*Proof.* Consider $t \geq 0$, $c_t \in \mathcal{C}_t(\delta)$, and let $\varphi_t := \langle c^* - c_t | \tilde{\pi}_t \rangle$. Recall that $a_t := \mathsf{F}\pi_t$. By Lemma C.2 and the Cauchy-Schwartz inequality, on the event $\{c^* \in \cap_{t \in \mathbb{N}} \mathsf{F}^{-1}\mathcal{C}_t(\delta)\}$, we have

$$|\varphi_t| \leq \beta_t(\delta) \|a_t\|_{(V_t^\lambda)^{-1}} \,,$$

while, by the Cauchy-Schwartz inequality, Assumption 3.1, and using the fact that $\mathsf{F}$ is an isometry on $L^2(\mathbb{R}^d; \varrho)$, we have

$$|\varphi_t| \leq \|\mathsf{F}\mathsf{R}c^* - \mathsf{F}\mathsf{R}c\|_{L^2(\mathbb{R}^d;\varrho)} \|a_t\|_{L^2(\mathbb{R}^d;\varrho)} \leq \|c^* - c\|_{L^2(\mathbb{R}^d;\varrho)} \pi_t(\mathbb{R}^d)\varrho(\mathbb{R}^d) \leq 2\bar{C}\,.$$

Combining yields

$$|\varphi_t| \leq \beta_t(\delta) \min\{\|a_t\|_{(V_t^\lambda)^{-1}}, 2\bar{C}\} = 2\bar{C}\beta_t(\delta)\left(\frac{1}{2\bar{C}}\|a_t\|_{(V_t^\lambda)^{-1}} \wedge 1\right)\,.$$

Squaring and applying the inequality $u \leq 2\log(1+u)$, which holds for $u \in [0,1]$, to the final term, yields

$$|\varphi_t|^2 \leq 8\bar{C}^2\beta_t(\delta)^2 \log\left(1 + \frac{1}{2\bar{C}}\|a_t\|_{(V_t^\lambda)^{-1}}\right)$$

and, summing up and using Jensen's inequality, we obtain

$$\sum_{t=1}^T \varphi_t \leq \sqrt{T \sum_{t=1}^T |\varphi_t|^2} \leq 2\bar{C}\beta_t(\delta)\sqrt{T \sum_{t=1}^T \log\left(1 + \frac{1}{2\bar{C}}\|a_t\|_{(V_t^\lambda)^{-1}}\right)}\,. \tag{21}$$

By definition of $M_T$ and $V_T^\lambda$, we have

$$\sum_{t=1}^T \log\left(1 + \frac{1}{2\bar{C}}\|a_t\|_{(V_t^\lambda)^{-1}}\right) = \log\left(\prod_{t=1}^T \left(1 + \frac{1}{2\bar{C}}\|a_t\|_{(V_t^\lambda)^{-1}}\right)\right)$$

$$= \log \det\left(\mathrm{Id} + \frac{1}{2\lambda\bar{C}} M_T(\mathrm{D}\Lambda)^{-1} M_T^*\right) \tag{22}$$

as wanted. $\qquad\square$

We are now in position to prove Theorem 4.1, which we restate for convenience below.

**Theorem 4.1.** *Under Assumptions 3.1 and 3.2, if $c^*$ is $\alpha$-Hölder on $supp(\mu) \times supp(\nu) \subset \mathbb{R}^d$, for some $\alpha \in (0,1]$, then for any $\delta > 0$, $\lambda > 0$, $\eta \in (0,1)$, and $T \in \mathbb{N}$, the regret of Algorithm 1 with $(\varepsilon_t)_{t \in \mathbb{N}} = (\eta t^{-\eta})_{t \in \mathbb{N}}$, denoted $\mathcal{A}$, satisfies*

$$\mathscr{R}_T(\mathcal{A}) \leq 2\bar{C}\beta_T(\delta)\sqrt{T \log \det\left(\mathrm{Id} + \frac{M_T(\mathrm{D}\Lambda)^{-1} M_T^*}{2\lambda\bar{C}}\right)}$$

$$+ \frac{\kappa\eta}{1-\eta}\left(T^{1-\eta}\log(T) + \frac{\eta}{2^\eta}\log(6)\right)$$

$$+ \sigma\sqrt{2T \log\left(\frac{2}{\delta}\right)}$$

*with probability at least $1 - \delta$, wherein $\kappa$ is a constant dependant on $(\mu, \nu, \alpha)$ (see Lemma C.3).*

*Proof.* Recall that we identify $\mathcal{A}$ with the $\mathbb{F}$-adapted process $\boldsymbol{\pi} := (\pi_t)_{t \in \mathbb{N}} \in \Pi(\mu, \nu)$ of transport plans played along the trajectory under consideration. The proof follows from a somewhat standard regret decomposition, except for three exceptions: 1) some parts must be done in the frequency domain, 2) due to the entropic regularisation, the optimism term is not directly eliminated and must be compensated with the approximation term, and 3) this approximation term must be controlled with Lemma C.3. These changes are direct consequences of the need for the algorithm to ensure a valid frequency domain representation of its actions, so that it can apply the optimism principle despite the actions being measure-valued.

Let us denote by $r_t$ the instantaneous regret at time $t$, that is,

$$r_t := C_t - \mathrm{Kant.}(\mu, \nu, c^*),$$

with, naturally, $\mathscr{R}_T(\mathcal{A}) := \sum_{t=1}^T r_t$. As usual, we first eliminate $\xi_t = C_t - \mathbb{E}[C_t | \mathcal{F}_{t-1}]$, by using a concentration argument and the fact that $(\xi_t)_{t \in \mathbb{N}}$ is a sub-Gaussian Martingale difference sequence, to show that, for any $\delta > 0$, we have

$$\mathbb{P}\left( \sum_{i=1}^T \xi_i \leq \sigma \sqrt{2T \log\left(\frac{2}{\delta}\right)} \right) \geq 1 - \frac{\delta}{2}.$$

by Lemma F.2.

Subsequently, we can continue the decomposition of $\bar{r}_t := \mathbb{E}[r_t | \mathcal{F}_{t-1}] = \langle c^* | \pi_t \rangle$. At this stage, we inject the entropic functional to apply the optimistic property of Algorithm 1, i.e., we decompose as

$$\begin{aligned}
\bar{r}_t &= \langle c^* | \pi_t \rangle - \mathrm{Kant.}(\mu, \nu, c^*) \\
&= \langle c^* | \pi_t \rangle - \mathrm{Ent.}(\mu, \nu, c^*, \varepsilon) + \mathrm{Ent.}(\mu, \nu, c^*, \varepsilon) - \mathrm{Kant.}(\mu, \nu, c^*) \\
&\leq \langle c^* | \pi_t \rangle - \mathrm{Ent.}(\mu, \nu, c^*, \varepsilon) + \kappa \varepsilon \log(\varepsilon^{-1})
\end{aligned}$$

for any $\varepsilon > 0$, by Lemma C.3. In particular, for $\varepsilon = \varepsilon_t$ as given by Theorem 4.1, we have

$$\sum_{t=1}^T \kappa \varepsilon_t \log(\varepsilon_t^{-1}) \leq \frac{\kappa \eta}{1 - \eta} \left( T^{1-\eta} \log(T) + \frac{\eta}{2^\eta} \log(6) \right).$$

by Lemma F.3.

Returning to the regret decomposition and $\langle c^* | \pi_t \rangle - \mathrm{Ent.}(\mu, \nu, c^*, \varepsilon)$, we recall, by optimism and since $\mathscr{H} > 0$, that

$$\langle \tilde{c}_t | \pi_t \rangle \leq \mathrm{Ent.}(\mu, \nu, \tilde{c}_t, \varepsilon_t) \leq \mathrm{Ent.}(\mu, \nu, c^*, \varepsilon_t) \text{ on } \mathcal{E}_t(\delta).$$

Recall that $\mathcal{E}_t(\delta)$ is the event on which the confidence sets are valid up to time $t$, see (19). Placing ourselves on the event $\cap_{t \in \mathbb{N}} \{c^* \in \mathsf{F}^{-1} \mathcal{C}_t(\delta)\} \supset \cap_{t \in \mathbb{N}} \mathcal{E}_t(\delta)$, which happens with probability at least $1 - \delta$ by Lemma C.2, we have

$$\langle c^* | \pi_t \rangle - \mathrm{Ent.}(\mu, \nu, c^*, \varepsilon_t) \leq \langle c^* - \tilde{c}_t | \tilde{\pi}_t \rangle.$$

This term is bounded by the width of the confidence sets, as shown in Lemma D.1. Gathering all the terms completes the bound. $\qquad\square$

### D.2. General trajectorial regret bounds

We now return to the general question of linear functionals and summarise a general working procedure to learn them in a stochastic bandit setting. This procedure follows the ideas developed for the Kantorovich problem: choice of a subspace which is Hilbertian, construction of a RLS estimator, and regularisation of the optimism step according to a barrier function. We identify that a $\Gamma$-convergence condition on the regularised problem is the essential ingredient in determining learnability. If this convergence is fast enough, as in optimal transport, then the resulting regret term will be of a lower-order relative to learning.

In order to generalise Algorithm 1 and Theorem 4.1, let us return to the general notation of Section 1, recall it briefly, and complete it. Let $\mathcal{S}$ be a set isomorphically embedded in a Hilbert space $(\mathcal{H}, \langle \cdot | \cdot \rangle_{\mathcal{H}})$ by an embedding $I : \mathcal{S} \to \mathcal{H}$. Let us also assume that there is a family of continuous, non-negative functionals $\{Q_\varepsilon : \varepsilon > 0\}$ on $\mathbb{A}$, whose effective domain is

a set $\mathcal{Q} \subset \mathbb{A}$ which can be embedded into $\mathcal{H}$ by an injection $\Upsilon : \mathcal{Q} \to \mathcal{H}$. Reusing the corresponding notation from the entropic transport functional, let $\Psi^\varepsilon : \mathcal{S} \times \mathbb{A} \to \mathbb{R} \cup \{+\infty\}$ be the functional defined by

$$\Psi^\varepsilon(\varsigma; a) := J^\varsigma(a) + Q_\varepsilon(a) \text{ for any } (\varsigma, a) \in \mathcal{S} \times \mathbb{A}.$$

Recall that, in the case of the BOT problem, we used $Q_\varepsilon := \varepsilon \mathscr{H}(\cdot | \varrho)$ and $\mathcal{Q} := \{\pi \in \mathscr{P}(\mathcal{X}) : \mathrm{d}\pi/\mathrm{d}\varrho \in L^2(\mathbb{R}^d; \varrho)\}$, with $I : \pi \in \mathcal{Q} \mapsto \mathrm{d}\phi/\mathrm{d}\varrho \in L^\infty(\mathbb{R}^d; \varrho)$. Other possible candidates include (positive) quadratic forms, barrier functions for $\mathcal{H}$ (e.g. $a \mapsto \varepsilon \|a\|_{\mathcal{H}}$) when $\mathcal{H} \subset \mathbb{A}$, and convex indicator functions.

Suppose we have a sequence of actions $(a_s)_{s \in \mathbb{N}} \subset \mathcal{Q}$, and a corresponding sequence of observed costs $(C_s)_{s \in \mathbb{N}} \subset \mathbb{R}$, $C_s := J^*(a_s) + \xi_s$ for each $s \in \mathbb{N}$ and with $(\xi_s)_{s \in \mathbb{N}}$ satisfying Assumption 3.2. Let $\Lambda : \mathcal{S} \to [0, +\infty)$ be a strongly convex and continuously Fréchet differentiable functional on $\mathcal{H}$ (e.g. $\Lambda = \|\cdot\|_{\mathcal{H}}^2$). Note that if $\mathcal{S}$ cannot be embedded in a Hilbert space, this assumption of Fréchet differentiability becomes non-trivial, as can be seen by the example $(\mathbb{R}^d, \|\cdot\|_1)$, where $\|\cdot\|_1^2$ is not Fréchet differentiable[7].

We can then define the least-squares estimator

$$\hat{\varsigma}_t := \underset{\varsigma \in \mathcal{S}}{\mathrm{argmin}} \sum_{s=1}^{t} |C_s - \langle I(\varsigma)|\Upsilon(a_s)\rangle_{\mathcal{H}}|^2 + \lambda \Lambda(I(\varsigma)),$$

where $\lambda > 0$ is a regularisation parameter. Concentration of least-squares residuals then shows an analogue to Lemma C.2 and thus the validity of the confidence sets

$$\tilde{\mathcal{C}}_t(\delta) := \left\{ \varsigma \in \mathcal{S} : \|I(\varsigma) - I(\hat{\varsigma}_t)\|_{D_t^\lambda} \leq \tilde{\beta}_t(\delta) \right\} \text{ for } t \in \mathbb{N}, \tag{23}$$

in which, $\|\cdot\|_{D_t^\lambda} := \sqrt{\langle \cdot | D_t^\lambda \cdot \rangle_{\mathcal{H}}}$ and

$$\tilde{\beta}_t(\delta) := \sigma \sqrt{\log\left(\frac{4\det(\mathrm{D}\Lambda + \lambda^{-1}M_t M_t^*)}{\delta^2}\right)} + \left(\frac{\lambda}{\|\mathrm{D}\Lambda\|_{\mathrm{op}}}\right)^{\frac{1}{2}} \|I(\varsigma^*)\|_{\mathcal{H}},$$

with $M_t : h \in \mathcal{H} \mapsto (\langle h|\Upsilon(a_s)\rangle_{\mathcal{H}})_{s=1}^t \in \mathbb{R}^t$ and $M_t^* : \mathbb{R}^t \to \mathcal{H}$ its adjoint.

Like in the BOT case, this restriction from $\mathbb{A}$ to $\mathcal{Q}$ comes at the expense of an approximation term which must be controlled as a function of $\varepsilon$ to ensure its accumulation does not dominate the regret. This is a highly ad-hoc problem, to which we will not attempt to provide a general solution, but rather provide a minimal assumption (Assumption D.2 below) and discuss sufficient conditions which may be investigated on a case-by-case basis.

**Assumption D.2.** There is a non-decreasing function $\varpi : (0, +\infty) \to (0, +\infty)$ satisfying $\lim_{\varepsilon \downarrow 0} \varpi(\varepsilon) = 0$ and a function $b : \mathcal{S} \to [0, +\infty)$ such that the family $\{Q_\varepsilon : \varepsilon > 0\}$ satisfies

$$\limsup_{\varepsilon \to 0} \frac{\inf_{a \in \mathbb{A}} \Psi^\varepsilon(\varsigma; a) - \inf_{a \in \mathbb{A}} J^\varsigma(a)}{\varpi(\varepsilon)} \leq b(\varsigma),$$

for any $\varsigma \in \mathcal{S}$.

Assumption D.2 asks for a rate of convergence of the minimum from $\Psi^\varepsilon$ to the minimum of $J^\varsigma$ as the penalisation $Q_\varepsilon$ vanishes. It is evident that this assumption is necessary in the sense that if there is no such rate function $\varpi$ or if $b(\varsigma^*) = +\infty$, then the algorithm must suffer linear regret in the worst case. However, this form of assumption does not appear to have been studied prior to this work, despite being a weaker version of the classical notion of $\Gamma$-convergence (a.k.a. epi-convergence), which is a standard tool in the study of variational problems.

On a metric space[8] $(M, d)$, a sequence of functionals $(F_n)_{n \in \mathbb{N}}$ from $M \to \mathbb{R} \cup \{\pm\infty\}$ $\Gamma$-converges to a functional $F : M \to \mathbb{R} \cup \{\pm\infty\}$ if, for every $x \in M$,

$$\liminf_{n \to \infty} F_n(x_n') \geq F(x) \text{ along any sequence } (x_n)_{n \in \mathbb{N}} \to x \text{ of points of } M$$

---

[7] In this case, it is still possible to construct confidence sets using the method of Abbasi-Yadkori et al. (2011, Thm. 3.18), provided a low-regret online linear regression oracle exists.

[8] This extends to non-metric topologies under technical assumptions whose discussion here is unnecessary.

and if there is sequence $(x'_n)_{n \in \mathbb{N}} \to x$ of points of $M$, such that $\limsup_{n \to \infty} F_n(x'_n) \leq F(x)$. In the context of BOT, $\Gamma$-convergence of $\Psi^\varepsilon(c^*; \cdot)$ to Kant.$(\mu, \nu, c^*)$ was established by Carlier et al. (2017), with convergence rates at the optimum given by Carlier et al. (2023).

Let us now turn to defining our algorithm and its regret. Define the sequence of actions $\mathcal{B} := (a_t)_{t \in \mathbb{N}}$ recursively by

$$a_{t+1} := \underset{a \in \mathbb{A}}{\operatorname{argmin}} \ \min_{\varsigma \in \tilde{\mathcal{C}}_t(\delta)} \ \Psi^{\varepsilon_t}(\varsigma; a), \text{ for } t \in \mathbb{N}, \tag{24}$$

with $\tilde{\mathcal{C}}_t(\delta)$ as in (23) and $\varepsilon_t := \varpi^{-1}(t^{-\eta})$ for any $t \in \mathbb{N}$ and with $\eta \in (0, 1)$. $\mathcal{B}$ is clearly an analogue to Algorithm 1. Finally, let us recall the definition of the regret in this context. For any $\mathbb{F}$-adapted sequence of actions $\mathcal{B} := (a_s)_{s \in \mathbb{N}} \subset \mathbb{A}$, its regret is

$$\mathscr{R}_T^J(\mathcal{B}) := \sum_{t=1}^T J^*(a_t) - \inf_{a \in \mathbb{A}} J^*(a) \text{ for } T \in \mathbb{N}.$$

Theorem D.3 provides a bound on the regret of the algorithm $\mathcal{B}$ under these general assumptions, generalising Theorem 4.1.

**Theorem D.3.** *Under the above assumptions, Assumption D.2, and assuming that $(\xi_t)_{t \in \mathbb{N}}$ is a sub-Gaussian Martingale difference sequence, the algorithm $\mathcal{B}$ satisfies*

$$\mathscr{R}_T^J(\mathcal{B}) \leq \sigma \sqrt{2T \log\left(\frac{2}{\delta}\right)} + 2\bar{C}\tilde{\beta}_T(\delta) \sqrt{T \log \det\left(\mathrm{Id} + \frac{1}{2\lambda\bar{C}} M_T(\mathrm{D}\Lambda)^{-1} M_T^*\right)}$$
$$+ \frac{\eta}{1-\eta}\left(T^{1-\eta}\log(T) + \frac{\eta}{2^\eta}\log(6)\right)$$

*with probability at least $1 - \delta$.*

*Proof.* It is clear that the proof follows the same lines as Theorem 4.1, we need only reproduce the regret decomposition in the general notation. First, let $\varsigma_t$ denote a minimiser in the interior minimum in (24), for any $t \in \mathbb{N}$. The instantaneous regret $r_t$ at time $t \in \mathbb{N}$ can be written as

$$r_t := J^*(a_t) - \inf_{a \in \mathbb{A}} J^*(a)$$
$$= J^*(a_t) - \Psi^\varepsilon(\varsigma_t; a_t) + \inf_{a \in \mathbb{A}} \Psi^\varepsilon(\varsigma_t; a) - \inf_{a \in \mathbb{A}} \Psi^\varepsilon(\varsigma^*, a) + \inf_{a \in \mathbb{A}} \Psi^\varepsilon(\varsigma^*, a) - \inf_{a \in \mathbb{A}} J^*(a), \tag{25}$$

by definition of $\mathcal{B}$. The first difference can be bound by positivity of $Q_\varepsilon$ by the confidence width term, i.e.,

$$J^*(a_t) - \Psi^\varepsilon(\varsigma_t; a_t) \leq \langle I(\varsigma^*) - I(\varsigma_t) | \Upsilon(a_t) \rangle_{\mathcal{H}}, .$$

where $\varsigma^* \in \mathcal{S}$ is the unknown true parameter of the problem, i.e. $J^{\varsigma^*} = J^*$.

By (24), the second difference in (25) is negative, while the third one is at most $\varpi(\varepsilon_t)$ by assumption. Since $\varepsilon_t := \varpi^{-1}(t^{-\eta})$ by definition of $\mathcal{B}$, we obtain the desired bound on the third term by applying Lemma F.3. Finally, analogues of Lemmas D.1 and C.2 complete the proof along the lines of the proof of Theorem 4.1. $\square$

While research on linear functionals of non-Hilbertian Banach spaces is dispersed across fields of application, let us close out this section by mentioning example problems which ought to be of interest to learning theorists.

Consider a discrete-time Markov decision process with state space $\mathcal{X}$ and action space $\mathbb{A}$, transition kernel $P : \mathcal{X} \times \mathbb{A} \to \mathscr{P}(\mathcal{X})$ and a cost function $c^* : \mathcal{X} \times \mathbb{A} \to [0, 1]$. Let $\Pi$ denote the set of admissible controls on this system, and for any $\pi \in \Pi$ let $(X_s^{\pi,x}, A_s^\pi)_{s \in \mathbb{N}}$ be the corresponding state-action random process from initial state $x \in \mathcal{X}$ at time $s = 0$. We can then define the functional

$$J^{c^*}(\pi; x) := \mathbb{E}\left[\sum_{t=0}^\infty \omega(t) c^*(X_t^\pi, A_t^\pi) \Big| X_0 = x\right], \text{ for } \pi \in \Pi, x \in \mathcal{X}, \tag{26}$$

for some discount function $\omega : \mathbb{N} \to [0, 1]$ such that $\sum_{t=0}^{\infty} \omega(t) < +\infty$. Classical choices of $\omega$ are $\omega(t) = \gamma^t$ for some $\gamma \in (0, 1)$ or $\omega(t) = \mathbb{1}_{\{t \leq H\}}$ for some $H \in \mathbb{N}$, we eschew discussion of the average-cost criterion for simplicity.

The optimal control problem requires finding an admissible policy that minimises $J^{c^*}(\cdot, x)$, i.e.

$$\pi_x^* \in \inf_{\pi \in \Pi} J^{c^*}(\pi; x). \tag{27}$$

In a (bandit) Reinforcement Learning (RL) problem, the cost function $c^*$ is unknown and optimal behaviour must be learned through interaction with the system with low regret. In the episodic feedback setting, the agent chooses a policy $\pi_t \in \Pi$ at the start of episode $t \in \mathbb{N}$, and then observes (at least) a reward signal which is a noisy observation of $J^{c^*}(\pi_t; x)$. It is readily apparent from (26) that $J^{c^*}(\cdot; x)$ is highly non-linear in general in $\pi$ and this problem appears much harder than a linear bandit.

Despite this, the problem (27) can be cast as a linear problem in a different variable. Indeed, let us define the occupation measure $\mu^{\pi,x} \in \mathscr{P}(\mathcal{X} \times \mathbb{A})$ of a policy $\pi \in \Pi$ from initial state $x \in \mathcal{X}$ by

$$\mu^{\pi,x}(B) := \mathbb{E}\left[\sum_{t=0}^{\infty} \omega(t) \mathbb{1}_B(X_t, A_t) \Big| X_0 = x\right], \text{ for } B \in \mathcal{B}(\mathcal{X} \times \mathbb{A}).$$

By rewriting the expectation in (26) as an integral with respect to $\mu^{\pi,x}$, we obtain

$$\pi_x^* \in \operatorname*{argmin}_{\pi \in \Pi} \int c^*(y, a) \mathrm{d}\mu^{\pi,x}(y, a).$$

This problem is now a linear problem on the space of measures, which suggests it can be approached with the methods of this paper, especially in view of existing work on entropy-regularized control by Neu et al. (2017). The pleasing prospect of this approach is isolating the complexity of the control system dynamics into the mapping $\pi \mapsto \mu^{\pi,x}$, separately from the statistical complexity of learning the cost function $c^*$.

Related linear problems include routing problems in continuous spaces (when transitions are deterministic), and the dynamic formulation of optimal transport if one imposes constraints on the distribution of $X_0$ and $X_T$ for some $T > 0$ (Benamou & Brenier, 2000).

## E. Non-parametric Estimation and Worst-case Regret

In this section, let us return to the question of worst-case regret in BOT to provide some complements to Section 5. Compared to Appendix D.2, we eschew the general formulation and return to the BOT problem as the method we develop is straightforward to extend to the general setting and in no way limited to BOT specifically. This approach relies on a basis decomposition in $\mathcal{H}$ (frequency domain), on which a decay rate assumption is placed (see Assumption 5.1). As the choice of basis is free, this assumption is highly flexible and can be adapted to a priori knowledge of the regularity of the cost function $c^*$.

We will begin by providing some background on functional approximation theory and related regularity conditions in Appendix E.1, before turning to introducing our functional regression procedure and some general lemmata in Appendix E.2, which leads us to general worst-case regret bounds. Finally, we touch on the case of finite-dimensional problems in Appendix E.3.

### E.1. Basis decomposition and regularity conditions

We provide below only a summary introduction to the field of functional approximation theory and Kolmogorov widths, as we feel needed for the following sections. One may refer to Lorentz (2005) or Pinkus (1985) for a more thorough introduction to this area of research. The key question underpinning this section is: what is the best choice of $(\phi_i)_{i \in \mathbb{N}}$ when we know $f^*$ lives in a specific regularity class?

Let us begin by assuming that we know that $f^* \in S$ for some known but generic set $S \subset L^2(\mathbb{R}; \varrho)$, say a specific regularity class. Kolmogorov (1991) studied and characterised the optimal choice of a basis to approximate elements of $S$, and thus the best $\zeta$ obtainable for Assumption 5.1. Specifically, this $\zeta$ is given by the sequence of Kolmogorov $n$-widths of $S$ in

$L^2(\mathbb{R}^d; \varrho)$, defined as

$$d_n(S) := \inf_{\substack{V_n \subset L^2(\mathbb{R}^d;\varrho) \\ \dim(V_n) \leq n}} \sup_{f \in S} \inf_{v \in V_n} \|f - v\|_{L^2(\mathbb{R}^d;\varrho)} \,. \tag{28}$$

Quite simply, $d_n(S)$ is the error of the best approximation of $S$ by a subspace of dimension $n$ in $L^2(\mathbb{R}^d; \varrho)$: it is optimal by definition.

Beyond characterising $\zeta$, Kolmogorov also studied the *extremal subspaces*, i.e. the minimisers in (28) for any value of $n \in \mathbb{N}$. For practical purposes, extremal subspaces (which are linear by definition) can be represented as the span of a collection of vectors $\phi_i \in L^2(\mathbb{R}^d; \varrho)$, $i \in \mathbb{N}$, which can be formed into an orthonormal basis. Thus, characterising the decay of the Kolmogorov $n$-width of $S$ and identifying a sequence of extremal subspaces is sufficient to specialise Assumption 5.1 to a specific choice of $S$.

In general, finding the extremal functions of the Kolmogorov $n$-widths is no easy task, but they are known for many regularity classes $S$. In the following, we will briefly discuss one example which concerns a well known family of regularity classes of $L^2(\mathbb{R}^d; \varrho)$: the Sobolev spaces.

To recall essential definitions (see e.g. (Brézis, 2011) for a comprehensive treatment), let us introduce some standard notation. Let $d \in \mathbb{N}$, any multi-index $\alpha \in \mathbb{N}^d$ defines the differential operator $\mathrm{D}^\alpha$ as $\partial_{x_1}^{\alpha_1} \cdots \partial_{x_d}^{\alpha_d}$. Let $|\alpha| := \|\alpha\|_1$ denote the order of the multi-index. For a domain $\Omega \subset \mathbb{R}^d$, we will denote by $H^m(\Omega)$ for $m \in \mathbb{N}$ the Sobolev space containing all $L^2(\Omega; \varrho)$ functions[9] which are $m$-times weakly differentiable and whose derivatives of order $n$ are also in $L^2(\Omega; \varrho)$. This space can be rewritten in several different ways, and in particular:

$$H^m(\Omega) := \left\{ f \in L^2(\Omega; \varrho) : \|\mathrm{D}^\alpha f\|_{L^2(\Omega;\varrho)} < +\infty \text{ if } |\alpha| \leq m \right\}.$$
$$= \left\{ f \in L^2(\Omega; \varrho) : \|f\|_{H^m(\Omega)} < +\infty \right\} \tag{29}$$

wherein

$$\|f\|_{H^m(\Omega)} := \sqrt{\sum_{|\alpha| \leq m} \|\mathrm{D}^\alpha f\|_{L^2(\Omega;\varrho)}^2} \,,$$

with the sum being over all multi-indices $\alpha$ of order at most $m$. In particular, one can show from (29) that $(H^m(\Omega), \|\cdot\|_{H^m(\Omega)})$ is a Hilbert space.

It is known from the work of Kolmogorov (1991) that the Kolmogorov $n$-width of $H^m([0,1])$ in $L^2([0,1])$ is of order $\mathcal{O}(n^{-m})$ asymptotically. Furthermore, he provided a characterisation of the extremal functions (and thus of the optimal basis) as the eigenfunctions of the differential operator $(-1)^m \mathrm{D}^{2m}$, or equivalently to the solutions to an ordinary differential equation of order $2m$. This formulation could be extended to the multi-dimensional case, but it would require more care to set up the differential operator. This connection to the spectrum of specific operators is reflected, e.g., in Hu et al. (2025).

It is important to note that, as we are learning in the frequency domain, Assumption 5.1 is not directly about the regularity of $c^*$ but rather of its Fourier transform $f^* := \mathsf{F}c^*(-\cdot)$. Understanding the regularity of elements of $\mathsf{F}^{-1}S$ for a given regularity class $S$ is a Harmonic analysis problem beyond the scope of this paper, so let us limit ourselves to discussing only the case of $S \subset H^m(\Omega)$ as an illustration. Precisely, let us introduce the Sobolev classes

$$\{W(m, L) : (m, L) \in \mathbb{N} \times [0, +\infty)\}$$

in which each Sobolev class $W(m, L)$ is defined as the ball of radius $L$ centred at 0 in $H^m(\Omega)$. These classes are a standard tool for characterising the difficulty of estimation in non-parametric statistics, see e.g. (Tsybakov, 2008; Wasserman, 2006). Let us finally introduce a regularity assumption on $c^*$ with Proposition E.1, which establishes that higher order integrability of the cost function $c^*$ directly translates to membership in a Sobolev class, and thus a Sobolev space, for its Fourier transform.

**Proposition E.1.** *Assume that $c^* \in L^2(\mathbb{R}^d; \varrho)$ satisfies the integrability (growth) condition:*

$$\int \left[(1 + \|z\|)^m c^*(z)\right]^2 \mathrm{d}\varrho(z) < M^2 \,, \tag{30}$$

---

[9] For ease of exposition, we gloss over the distinction between functions and equivalence classes here.

*for some $M > 0$ and that $supp(\mu \otimes \nu) \subset \Omega$ for some bounded open domain $\Omega \subset \mathbb{R}^d$. Then, we have $\mathsf{F}c^* \in W(m, CM) \subset H^m(\Omega)$ for some constant $C > 0$.*

*Proof.* We can use the Fourier transform's effect on differentials to write

$$\|D^\alpha \mathsf{F}c^*\|^2_{L^2(\mathbb{R}^d;\varrho)} \leq C^2 \int \left[(1 + \|z\|)^m \left|\mathsf{F}^{-1}\mathsf{F}c^*(z)\right|\right]^2 d\varrho(z) < C^2 M^2 \,,$$

for some constant $C > 0$ and for every multi-index $\alpha$. Consequently, $\mathsf{F}c^* \in W(m, CM)$, as wanted. $\qquad\square$

We begin by setting the stage with a fixed order (i.e. $n$ independent of $t$) methodology. Later, we will derive regret guarantees when $n$ is allowed to grow with $t$ in order to control the approximation error.

In summary, specific regularity conditions on $c^*$ can be incorporated into the choice of the basis $(\phi_i)_{i\in\mathbb{N}}$, yielding an appropriate approximation error $\zeta$, with the Kolmogorov $n$-widths, i.e. $\zeta \equiv d.(S)$, being the best possible. The choice of basis can also come from the regularity of the marginals, for example Hermite polynomials are a natural choice for Gaussian marginals. In particular, the existence of a parametric model allows us to choose the basis so that $\zeta \equiv 1$ after some $N \in \mathbb{N}$, i.e. $\gamma_i^* = 0$ for $i > N$. This flexibility sets functional regression apart from naive discretisations of the problem, which suffer from the large covering numbers of function classes.

### E.2. Functional regression with basis truncation

The section is dedicated to the design of confidence sets for the functional regression problem with basis truncation, and to establishing their uniform validity when the truncation order $n$ is allowed to vary with time. We begin by giving the simple proof of Lemma 5.2, which controls the approximation error induced by truncating the basis expansion of $f^*$ at order $n$.

**Lemma 5.2.** *Let $(\phi_i)_{i\in\mathbb{N}}$ be an orthonormal basis of $L^2(\mathbb{R}^d;\varrho)$, and let $f \in L^2(\mathbb{R}^d;\varrho)$ with $f := \sum_{i=0}^\infty \gamma_i \phi_i$. Then, for every $g \in L^2(\mathbb{R}^d;\varrho)$, we have*

$$\left|\langle f - f|_n | g\rangle_{L^2(\mathbb{R}^d;\varrho)}\right| \leq \|g\|_{L^2(\mathbb{R}^d;\varrho)} \sqrt{\sum_{i=n+1}^{+\infty} |\gamma_i|^2}.$$

*Proof.* Write $g = \sum_{i=1}^\infty \zeta_i \phi_i$ for some $\zeta \in \ell_2(\mathbb{R})$. Then, we have

$$\langle f - f|_n | g\rangle_{L^2(\mathbb{R}^d;\varrho)} = \left\langle \sum_{i=n+1}^\infty \gamma_i \phi_i \Big| g \right\rangle_{L^2(\mathbb{R}^d;\varrho)},$$

so that by the Cauchy-Schwarz inequality and orthonormality of $(\phi_i)_{i\in\mathbb{N}}$ we get

$$\left|\langle f - f|_n | g\rangle_{L^2(\mathbb{R}^d;\varrho)}\right| \leq \|g\|_{L^2(\mathbb{R}^d;\varrho)} \sqrt{\sum_{i=n+1}^\infty |\gamma_i|^2},$$

as wanted. $\qquad\square$

For any fixed $n \in \mathbb{N}$, one can approximately regress $C_t$ against $a_t$ up to order $n$ by solving the $n$-dimensional Regularised Least-Squares (RLS) problem

$$\hat{\gamma}_t^{n,\lambda} := \operatorname*{argmin}_{\gamma\in\mathbb{R}^n} \sum_{s=1}^t \left\| C_s - \sum_{i=1}^n \gamma_i \vartheta_i^{(s)} \right\|_2^2 + \lambda \Lambda_n(\gamma) \,, \tag{31}$$

in which $\Lambda_n : \mathbb{R}^n \to [0, +\infty)$ is a strictly convex continuously Fréchet-differentiable regulariser such that its Fréchet derivative $D\Lambda_n$ satisfies

$$\frac{1}{M_{\Lambda_n}} \operatorname{Id} \preceq D\Lambda_n \preceq M_{\Lambda_n} \operatorname{Id} \,.$$

For clarity, let $\vartheta^{(s,n)}$ denote the truncation of $\vartheta^{(s)} \in \mathbb{R}^{\mathbb{N}}$ at order $n$, so that $\vartheta^{(s,n)} \in \mathbb{R}^n$ and $\vartheta_i^{(s,n)} = \vartheta_i^{(s)}$ for all $i \in [n]$. Following the standard arguments for online linear regression of Section 2, one can construct the confidence sets (see Abbasi-Yadkori et al., 2011)

$$\tilde{\mathcal{C}}_t^n(\delta) := \left\{ \gamma \in \mathbb{R}^n : \left\| \gamma - \hat{\gamma}_t^{n,\lambda} \right\|_{\tilde{V}_t^{\lambda,n}} \leq \tilde{\beta}_{t,n}(\delta) \right\} , \tag{32}$$

in which $\tilde{V}_t^{\lambda,n} := \lambda \mathrm{D}\Lambda_n + \sum_{s=1}^t \vartheta^{(s,n)} \vartheta^{(s,n)\top}$ and

$$\tilde{\beta}_t^n(\delta) := \sigma \sqrt{\log \left( \frac{4 \det \left( \mathrm{D}\Lambda_n + \lambda^{-1} \sum_{s=1}^t \vartheta^{(s,n)} \vartheta^{(s,n)\top} \right)}{\delta^2} \right)} + \left( \frac{\lambda}{\|\mathrm{D}\Lambda_n\|_{\mathrm{op}}} \right)^{\frac{1}{2}} \bar{C} . \tag{33}$$

Notice that $C > \|c^*\|_{L^2(\mathbb{R}^d;\varrho)}$ implies that $C > \|\gamma^*\|_{\ell_2(\mathbb{R})}$ by definition of $(\phi_i)_{i \in \mathbb{N}}$, so that $\bar{C}$ is a valid upper bound on $\|\gamma^*\|_{\ell_2(\mathbb{R})}$.

While the validity of $(\tilde{\mathcal{C}}_t^n(\delta))_{t \in \mathbb{N}}$ follows from standard arguments, the validity of $(\tilde{\mathcal{C}}_t^{n_t}(\delta))_{t \in \mathbb{N}}$ for any non-decreasing sequence $(n_t)_{t \in \mathbb{N}} \subset \mathbb{N}$ is unclear. Before proceeding to the design of an optimistic algorithm let us establish this fact in Lemma E.2. To state the lemma, let

$$\tilde{\mathcal{E}}_t^n(\delta) := \left\{ \left\| \gamma^* - \hat{\gamma}_t^{n,\lambda} \right\|_{\tilde{V}_t^\lambda} \leq \tilde{\beta}_t^n(\delta) \right\} \quad \text{for} \quad (t,n) \in \mathbb{N}^2 .$$

**Lemma E.2.** *Under Assumptions 3.1 and 3.2, let $(n_t)_{t \in \mathbb{N}} \subset \mathbb{N}$ be a non-decreasing sequence. Then*

$$\mathbb{P} \left( \bigcap_{t=1}^\infty \tilde{\mathcal{E}}_t^{n_t}(\delta) \right) \geq 1 - \frac{\delta}{2} .$$

*Proof.* The proof only requires diagonalisation of the standard stopping time construction. For $(\delta, t) \in (0,1) \times \mathbb{N}$, on the filtered probability space $(\Omega, \mathcal{F}_\infty, \mathbb{F}, \mathbb{P})$ define

$$B_t(\delta) := \left\{ \omega \in \Omega : \left\| \gamma^* - \hat{\gamma}_t^{n_t,\lambda} \right\|_{\tilde{V}_t^{\lambda,n_t}} \leq \tilde{\beta}_{t,n_t}(\delta) \right\} \overset{\text{a.s.}}{=} \left\{ \omega \in \Omega : c^*|_{n_t} \notin \tilde{\mathcal{C}}_t^{n_t}(\delta) \right\} ,$$

be the $t^{\text{th}}$ "bad event", and let $\tau_\delta : \omega \in \Omega \to \inf\{t \in \mathbb{N} : \omega \in B_t(\delta)\}$, which is a stopping time. We have

$$\{\tau < +\infty\} = \bigcup_{t \in \mathbb{N}} B_t(\delta) .$$

By construction, in the classical manner:

$$\mathbb{P} \left( \bigcup_{t \in \mathbb{N}} B_t(\delta) \right) = \mathbb{P}(\tau < +\infty, B_t(\delta)) \leq \mathbb{P} \left( \tilde{\mathcal{E}}_t^{n_t}(\delta) \right) \leq \frac{\delta}{2} .$$

$\square$

Applying this learning methodology to Algorithm 1 in place of the infinite-dimensional RLS, and with the optimistic choice of belief-action pairs

$$(\tilde{\pi}_{t+1}, \tilde{\gamma}_{t+1}^{n_t}) \in \underset{\substack{\pi \in \Pi(\mu,\nu) \\ \gamma \in \tilde{\mathcal{C}}_t^{n_t}(\delta)}}{\operatorname{argmin}} \Psi_{\mu,\nu}^\varepsilon \left( \mu, \nu, \sum_{i=1}^{n_t} \gamma_i \phi_i, \varepsilon \right) \tag{34}$$

yields Algorithm 2.

Before moving on to the regret analysis, we must adapt the proof of Lemma D.1 to the case of varying $n_t$ in order to control the width term in the regret bound. Indeed, the steps summed up in (21) are no longer homogenous, and, in particular, (22) is no longer valid. This issue is patched up by Lemma E.3.

**Lemma E.3.** *Under Assumptions 3.1 and 3.2, if $\Lambda_n := \frac{1}{2} \left\| \cdot \right\|_2^2$ with the norm being on $\mathbb{R}^n$, then on the event $\bigcap_{t=1}^{\infty} \tilde{\mathcal{E}}_t^{n_t}(\delta)$ of Lemma E.2, for any non-decreasing sequence $(n_t)_{t \in \mathbb{N}} \subset \mathbb{N}$ and any $T \in \mathbb{N}$, we have*

$$\sum_{t=1}^{T} \langle c^* |_{n_t} - \tilde{c}_t^{n_t} | \tilde{\pi}_t \rangle \le 2C\sigma \left( \sqrt{2n_T \log \left( \frac{\lambda^{-1} + \frac{T\bar{C}^2}{n_T}}{\delta} \right)} + \sqrt{\lambda}\bar{C} \right) \sqrt{n_T T \log \left( 1 + \frac{T}{n_T \bar{C}^2} \right)}.$$

*Proof.* Recall the notation of Lemma D.1, which adapts to $\varphi_t := \langle c^* |_{n_t} - c_t^{n_t} | \tilde{\pi}_t \rangle$ for $t \in \mathbb{N}$ and $\tilde{c}_t^{n_t} := \sum_{i=1}^{n_t} \tilde{\gamma}_{t,i}^{n_t} \phi_i$. The proof of Lemma D.1 yields

$$\sum_{t=1}^{T} \varphi_t \le 2C\beta_{T,n_T}(\delta) \sqrt{T \sum_{t=1}^{T} \log \left( 1 + \frac{1}{2\bar{C}} \left\| \vartheta^{(t,n_t)} \right\|_{(\tilde{D}_t^{\lambda,n_t})^{-1}} \right)}.$$

First, one can bound $\tilde{\beta}_{T,n_T}(\delta)$ by the matrix-determinant lemma as in (Abbasi-Yadkori, 2012, Lemma E.3). To apply this lemma to the logarithmic second term, we must first adapt it to the desired form by conforming the vectors $\vartheta^{(t,n_t)}$. To do so, let us define the block matrices

$$Z_t := \begin{pmatrix} (\tilde{V}_t^{\lambda,n_t})^{-1} & \mathbf{0} \\ \mathbf{0} & \mathbf{0} \end{pmatrix} \qquad \text{for } t \in \mathbb{N},$$

so that we may use the rank one update formula to write

$$\prod_{t=1}^{T} \left( 1 + \frac{1}{2\bar{C}} \left\| \vartheta^{(t,n_t)} \right\|_{(\tilde{D}_t^{\lambda,n_t})^{-1}} \right) = \frac{\det \left( \mathrm{D}\Lambda_n + \sum_{t=1}^{T} \vartheta^{(t,n_t)} Z_t \vartheta^{(t,n_t)\top} \right)}{\det(\mathrm{D}\Lambda_n)}.$$

Finally, taking $\Lambda_n = \frac{1}{2} \left\| \cdot \right\|_2^2$, as given, we can bound the determinant of the numerator by

$$\det \left( \mathrm{D}\Lambda_n + \sum_{t=1}^{T} \vartheta^{(t,n_t)} Z_t \vartheta^{(t,n_t)\top} \right) \le \left( 1 + \frac{T}{n_T \bar{C}^2} \right)^{n_T}.$$

$\square$

Having established the technical lemmata, we now turn to the regret guarantees of the varying order basis truncation version of Algorithm 2. In particular, recall Assumption 5.1 to give a quantification of the regularity of $c^*$, which in turn will allow us to tune $(n_t)_{t \in \mathbb{N}}$ to obtain the best possible regret bounds in Theorem 5.3.

**Theorem 5.3.** *Assume Assumptions 3.1 to 3.2 and $\zeta(n) = 1 - n^{-q}$ for some $q > 0$. For any $\delta \in (0,1)$, $\lambda > 0$, $\varepsilon > 0$, let $\tilde{\mathcal{A}}$ denote Algorithm 2 with $(n_t)_{t \in \mathbb{N}} = (\lceil t^{\frac{1}{q+2}} \rceil)_{t \in \mathbb{N}}$, $\Lambda_n = \frac{1}{2} \left\| \cdot \right\|_2^2$, for all $n \in \mathbb{N}$, and $(\varepsilon_t)_{t \in \mathbb{N}} = (\eta t^{-\eta})_{t \in \mathbb{N}}$. For any $T \in \mathbb{N}$, the following regret bound holds:*

$$\mathscr{R}_T(\tilde{\mathcal{A}}) \le \bar{C} \left( 1 + \frac{2(q+2)}{q} T^{\frac{q+4}{2q+4}} \right) + \kappa(1 + \sqrt{T} \log(T))$$

$$+ 2\bar{C}\sigma T^{\frac{q+4}{2q+4}} \left( \sqrt{2 \log \left( \frac{\lambda^{-1} + 2T^{\frac{q+1}{q+2}} \bar{C}^2}{\delta} \right)} + \sqrt{\lambda}\bar{C} \right)$$

$$\times \sqrt{\log \left( 1 + \frac{2T^{\frac{q+1}{q+2}}}{\bar{C}^2} \right)} + \sigma \sqrt{2T \log \left( \frac{2}{\delta} \right)}.$$

*Proof.* The proof requires only a couple of steps from the one of Theorem 4.1, let us sketch them briefly. First, we modify the decomposition of the instant regret to include the truncation error:

$$\begin{aligned}
\bar{r}_t &= \langle c^* | \pi_t \rangle - \mathrm{Kant.}(\mu, \nu, c^*) \\
&= \langle c^* - c^* |_{n_t} | \pi_t \rangle + \langle c^* |_{n_t} | \pi_t \rangle - \mathrm{Kant.}(\mu, \nu, c^* |_{n_t}) \\
&\quad + \mathrm{Kant.}(\mu, \nu, c^* |_{n_t}) - \mathrm{Kant.}(\mu, \nu, c^*).
\end{aligned}$$

Second, we bound the resulting approximation error terms using Lemma 5.2 twice, which implies that

$$|\langle c^* - c^*|_{n_t}|\pi\rangle| \leq \sqrt{\sum_{k=n_t+1}^{+\infty} |\gamma_i^*|^2},$$

for any $\pi \in \Pi(\mu, \nu)$. Summing over $t \in \mathbb{N}$, one obtains

$$\sum_{t=1}^{T} |\langle c^* - c^*|_{n_t}|\pi_t\rangle| \leq \sum_{t=1}^{T} \sqrt{\sum_{i=n_t+1}^{+\infty} |\gamma_i^*|^2}. \tag{35}$$

By Assumption 5.1, for any $n \in \mathbb{N}$, we have

$$\sum_{i=n_t+1}^{\infty} |\gamma_i^*|^2 = \|c^*\|_{L^2(\mathbb{R}^d;\varrho)}^2 - \sum_{i=1}^{n_t} |\gamma_i^*|^2 \leq \|c^*\|_{L^2(\mathbb{R}^d;\varrho)}^2 (1 - \zeta(n_t))$$

so that for any $u > 0$, the choice $n_t := \lceil \zeta^{-1}((1 - t^{-u})) \rceil = \lceil t^{\frac{u}{q}} \rceil$ $(q > 0)$ yields

$$\sqrt{\sum_{i=n_t+1}^{\infty} |\gamma_i^*|^2} \leq \|c^*\|_{L^2(\mathbb{R}^d;\varrho)} t^{-\frac{u}{2}}. \tag{36}$$

This follows from the fact that $\zeta$ can be made a bijection of $\mathbb{R}_+ \to (0, 1]$, and that $\zeta$ is increasing. Injecting (36) into (35) yields

$$\sum_{t=1}^{T} |\langle c^* - c^*|_{n_t}|\pi_t\rangle| \leq \|c^*\|_{L^2(\mathbb{R}^d;\varrho)} \left(1 + 2\frac{T^{1-\frac{u}{2}}}{u}\right).$$

Thus, the cumulated expected regret can be bounded by

$$\sum_{t=1}^{T} \bar{r}_t \leq 2C \left(1 + 2\frac{T^{1-\frac{u}{2}}}{u}\right) + \sum_{t=1}^{T} (\langle c^*|_{n_t}|\pi_t\rangle - \text{Kant.}(\mu, \nu, c^*|_{n_t})).$$

Notice that the second term is the regret of Algorithm 2 against $c^*|_{n_t}$, which is easily controlled using optimistic analysis via Lemma E.2 and finally Lemma E.3 for $n_T := \lceil T^{\frac{u}{q}} \rceil \leq 2T^{u/q}$. This yields a bound of order $\mathcal{O}(T^{\frac{1}{2}+\frac{u}{q}})$. Setting $u = \frac{q}{q+2}$ yields the statement of the theorem. $\square$

### E.3. Finite-dimensional (parametric) problems

Consider an OT problem in which the measures $\mu$ and $\nu$ are supported on $K$ and $K'$ loci respectively (this is known as a *matching* problem). Let $\{x_1, \ldots, x_K\} = \text{supp}(\mu)$ and $\{y_1, \ldots, y_{K'}\} = \text{supp}(\nu)$ denote these loci. We can let $c^*$ assume arbitrarily values outside of $\mathcal{X} = \{(x_i, y_j) : (i, j) \in [K] \times [K']\}$ without loss of generality. Let $\epsilon < \inf\{\|u - v\| : (u, v) \in \mathcal{X}^2, u \neq v\}$ and define the functions

$$\phi_{i,j} := \frac{6}{\pi \epsilon^3} \mathbb{1}_{\{B_2((x_i, y_j)^\top, \epsilon/2)\}} \quad \text{for } (i, j) \in [K] \times [K'].$$

Re-indexing the functions by $k \in [K \times K']$, and adding suitable functions for $k > KK'$, we obtain an orthonormal basis $(\phi_k)_{k \in \mathbb{N}}$ of $L^2(\mathbb{R}^d; \varrho)$, in which $c^* := \sum_{k=1}^{KK'} \gamma_k^* \phi_k$. Consequently, we can apply Proposition 5.5 with $N = KK'$ to obtain a regret bound of $\tilde{\mathcal{O}}(CKK'\sqrt{T})$ for the learning problem. This implies the bound is tight with those obtained by representing this problem as a linear bandit in dimension $KK'$.

Alternatively, consider that there is a parametric model for $c^*$, i.e. there is $\theta^* \in \mathbb{R}^p$ such that

$$c^*(x, y) = \sum_{i=1}^{p} \theta_i^* \Phi_i(x, y),$$

for some embedding function $\Phi : \mathbb{R}^d \times \mathbb{R}^d \to \mathbb{R}^p$. When the embedding function is known, one can construct a basis through the Gram-Schmidt process. Let $\phi_1 := \Phi_1 / \|\Phi_1\|_{L^2(\mathbb{R}^d;\varrho)}$, and for $i \leq p$, define $S_i := \{\phi_k : k < i\}^\perp$ the orthogonal complement of the sequence this far. Now, repeatedly project (the projection operator being denoted by $P_{S_i}$) the feature dimensions onto $S_i$ to construct $\phi_i := P_{S_i}\Phi_i / \|P_{S_i}\Phi_i\|_{L^2(\mathbb{R}^d;\varrho)}$. For $i > p$, take any orthonormal basis of $S_p$ to complete the basis, it will not be used anyway. Consequently, we can also apply Proposition 5.5 with $N = p$ to obtain a regret bound of $\tilde{\mathcal{O}}(p\sqrt{T})$ for the learning problem. This is also tight with the corresponding linear bandit.

These results are summarised in Proposition 5.5, but notice that higher order polynomial models can be readily considered as well, such as quadratic costs

$$c^*(x, y) = \Phi(x, y)^\top \Theta^* \Phi(x, y) \,,$$

for $\Theta^* \in \mathbb{R}^{p \times p}$, by simply reparametrising it as a linear model in dimension $p^2$ and applying the same construction. Many other models can be considered in this manner, and would benefit from further specialised investigation.

# F. Miscellaneous Lemmata and Proofs

We regroup in this appendix some auxiliary results: a concentration argument in Appendix F.1, and a commonly used summation identity in Appendix F.2.

## F.1. Sub-Gaussian analysis

**Definition F.1.** A random variable $\xi : \Omega \to \mathbb{R}$ is $\sigma^2$-sub-Gaussian if

$$\mathbb{E}\left[\exp\left(t\xi\right)\right] \leq \exp\left(\frac{\sigma^2 t^2}{2}\right) \quad \text{for any } t \in \mathbb{R} \,.$$

A stochastic process $(\xi_i)_{i \in \mathbb{N}} : \Omega \to \mathbb{R}^\mathbb{N}$ is $\sigma^2$-conditionally sub-Gaussian if

$$\mathbb{E}\left[\exp\left(t\xi_i\right)\Big|\sigma((\xi_j)_{j<i})\right] \leq \exp\left(\frac{\sigma^2 t^2}{2}\right) \quad \text{for all } i \in \mathbb{N} \text{ and any } t \in \mathbb{R} \,.$$

**Lemma F.2.** *Let $(\xi_i)_{i \in \mathbb{N}}$ be a $\sigma^2$-conditionally sub-Gaussian process,*

$$\mathbb{P}\left(\sum_{i=1}^n \xi_i \geq \sigma\sqrt{2n\log\left(\frac{1}{\delta}\right)}\right) \leq \delta \quad \text{for any } (n, \delta) \in \mathbb{N} \times (0, 1) \,.$$

*Proof.* The proof follows Chernoff's method, by exponentiating $\sum_{i=1}^n \xi_i$ using $x \mapsto e^{tx}$, applying Markov's inequality, the tower rule accompanied by conditional sub-Gaussianity, and finally optimising the bound over the parameter $t > 0$. $\square$

## F.2. A common summation identity

**Lemma F.3.** *For $\eta \in (0, 1)$, let $\phi : u \in (0, +\infty) \mapsto \eta u^{-\eta}\log(u) \in \mathbb{R}_+^*$, then for any $N \in \mathbb{N}$,*

$$\sum_{u=1}^N \phi(u) \leq \frac{\eta}{1-\eta}N^{1-\eta}\log(N) + \frac{\eta}{2^\eta}\log(6) \,.$$

*In particular, if $\eta = 1/2$, then*

$$\sum_{u=1}^N \phi(u) \leq \sqrt{N}\log(N) + \frac{1}{2\sqrt{2}}\log(6) \,.$$

*Proof.* Notice that $\phi$ is differentiable, with $\phi'(u) = \eta u^{-(1+\eta)}(1 - \eta\log(u))$, so that it is decreasing on $(e^{1/\eta}, +\infty)$. Since $\sup_{\eta>1} e^{1/\eta} = e < 3$, comparison between the sum and the integral of $\phi$ yields

$$\sum_{u=1}^N \phi(u) \leq \phi(1) + \phi(2) + \phi(3) + \int_3^N \phi(u)\mathrm{d}u \,.$$

The remaining integral can be computed by parts, for $(a, b) \in \mathbb{R}_+^2$, $a < b$,

$$\int_a^b \phi(u)\mathrm{d}u = \frac{\eta}{1-\eta}\left(\left[u^{1-\eta}\log(u)\right]_a^b - \int_a^b u^{-\eta}\mathrm{d}u\right) \tag{37}$$

$$= \frac{\eta}{1-\eta}\left(\left[u^{1-\eta}\left(\log(u) - \frac{1}{\eta-1}\right)\right]_a^b\right) \tag{38}$$

$$\leq \frac{\eta}{1-\eta}b^{1-\eta}\log(b) \tag{39}$$

for every $\eta \in (0, 1)$. Computing yields $\phi(1) = 0$, $\phi(2) = \eta 2^{-\eta}\log(2)$, and $\phi(3) = \eta 3^{-\eta}\log(3)$, so that $\phi(1) + \phi(2) + \phi(3) \leq \eta 2^{-\eta}\log(6)$. Combining the results yields the desired inequality. $\qquad\square$

## G. Discussion of some Open Problems

We now provide some complements to the discussion of open problems in Section 6 which were omitted from the main text for brevity.

### G.1. Unknown marginals

Two possible directions appear to resolve this issue: one at the level of bandit design, and one at the level of numerical optimal transport. The former revolves around the idea of incorporating action-set violations to regret analysis, the latter around the idea of modifying Sinkhorn's algorithm to produce valid primal iterates at each step, e.g. by projecting onto $\Pi(\mu, \nu)$.

The question of violating action sets has been posed before in Bandit Theory and has also arisen in practical use-cases in Reinforcement Learning, see (Seurin et al., 2020). It is a staple topic in the context of fairness, see e.g. (Joseph et al., 2016) and of contextual bandits (including linear stochastic bandits) in which various other types constraint have also been considered, see e.g. (Liu et al., 2024). These types of constraints typically, in effect, disable certain arms at certain times, a generic setting which has been considered as well, e.g. by (Kleinberg et al., 2010; Abensur et al., 2019).

These works adopt a range of strategies to formulate the problem in a meaningful way, but their perspectives don't really fit with the real challenge we have with the OT problem. The problem isn't so much that the constraints placed on the action set are complicated: $\Pi(\mu, \nu)$ is a convex, compact set defined by linear inequalities. The problem arises entirely from the facts that $\Pi(\mu, \nu)$ is infinite-dimensional, and that it is a subspace of $\mathscr{P}(\mathcal{X})$, whose geometry is far from straightforward.

A preliminary exploration of this topic would likely require a taxonomy of the different possible violations of $\Pi(\mu, \nu)$. Indeed, $\pi_t$ could violate one or both marginal constraints, or it could even fail to be a probability measure through the total mass or positivity conditions. It appears likely that these will have quite different impacts both on the problem's geometry and on practical usefulness. Thereafter, one might consider whether guaranteeing finitely many violations, as (Liu et al., 2024) do, or developing a penalised regret is more appropriate.

Alternatively, one could design an algorithm which optimises the entropic or Kantorovich problems while staying within the constraint set $\Pi(\mu, \nu)$ (either for all time, or once it reaches a desired precision). On the one hand, there are finite-dimensional intuitions for this to work as Sinkhorn's algorithm can be viewed as a form of gradient descent (Léger, 2021), which could be projected onto $\Pi(\mu, \nu)$ (which is convex and compact). On the other hand, the geometry of $\Pi(\mu, \nu)$ as an infinite-dimensional probability space is likely to make rigorously doing so (and deriving convergence rates) quite arduous work. Nevertheless, this presents an inherent interest from the point of view of optimisation theory.

### G.2. Extension to the Monge problem

The *Monge* optimal transport problem associated to $(\mu, \nu, c)$ is

$$\mathrm{Monge}(\mu, \nu, c) := \inf_{T \in \mathscr{T}} \int c(x, T(x))\mathrm{d}\mu(x), \tag{40}$$

in which $\mathscr{T}$ is the set of all $\mu$-measurable maps $T : \mathcal{M}_\mu \to \mathcal{M}_\nu$ such that $\mu(T^{-1}(\cdot)) = \nu$. Chronologically, this is in fact the original formulation of the OT problem (Monge, 1781).

The Monge problem is best approached through finite-dimensional practical applications such as *matchings* of students to universities, employees to employers, etc. The requirement that the map $T$ be a function imposes an *indivisibility* of the mass $T$ moves from $\mu$ to $\nu$ (i.e. one university per student). This makes the resolution of the problem much more difficult. For example, if $\mu$ and $\nu$ each have two atoms with weights $(1/2, 1/2)$ and $(1/3, 2/3)$ respectively, then $\mathcal{T} = \emptyset$, meaning $\mathrm{Monge}(\mu, \nu, \cdot) \equiv +\infty$, and the problem is never solvable.

If $\mu, \nu$ are non-atomic, $\mathrm{Monge}(\mu, \nu, c)$ can be interpreted as the cheapest way (w.r.t. $c$) to transport a $\mu$-shaped pile of infinitesimally small things into a $\nu$-shaped one, but its geometry remains complicated. The Kantorovich relaxation drastically simplified the geometry of the problem and remains one of the most effective tools to approach the Monge problem, which is why it is accepted as the standard in modern OT theory.

Note that the relaxation from $\mathrm{Monge}(\mu, \nu, c)$ to $\mathrm{Kant.}(\mu, \nu, c)$ is known to be exact in some cases, such as $c = \|\cdot - \cdot\|^2 / 2$ with $\mathcal{M}_\mu = \mathcal{M}_\nu = \mathbb{R}^d$, $(\mu, \nu)$ having second-order moments and $\mu$ being absolutely continuous w.r.t. the Lebesgue measure (Ambrosio et al., 2021, Thm. 5.2). See also Villani (2009, Thm. 5.30) for weaker conditions. But it is also known (e.g. via the above example) that this relaxation is not without loss.

If we want to learn a Monge problem, we must, of course, make sufficient assumptions for it to be solvable, but more importantly we must face the issue that (40) is now a non-linear functional and that $\mathcal{T}$ is not as docile a set as $\Pi(\mu, \nu)$. Here, the recent work in statistical optimal transport on learning Monge maps (i.e. the solutions to (40)) is highly relevant, see e.g. Chewi et al. (2024, Ch. 3) or the appropriate paragraph in Appendix H below. Though once again most work focuses on the batch sampling of marginals, not on online learning. This line of work would appear to also require more general results about the learning of minima of non-linear functionals, which are not yet available in the literature. Overall, it remains unclear if the Monge problem is on a similar or different level of difficulty to the Kantorovich problem and this provides an interesting avenue to verify the necessity of linearity for low regret learning.

Beyond these statistical issues, one should also expect the problems of effective optimisation from Appendix G.1 to return with a vengeance as the Monge problem is a fully non-linear problem unlike the Kantorovich problem which is an (infinite-dimensional) linear program.

## H. Bibliographical Complements

An excellent detailed history of the development of OT as a mathematical theory, replete with bibliographical notes, can be found in Villani (2003, Ch. 3). Summarising this field's venerable history further would be of little value. Instead, we will expand on relevant research specifically about *learning* optimal transport problems, which is generally grouped under the term "statistical optimal transport". We touch on key aspects of the literature below, and refer to the monograph of Chewi et al. (2024), for a deeper longitudinal overview.

**Estimation of Wasserstein distances**    One of the most important contributions of optimal transport is a family of useful distances between probability measures: the Wasserstein metrics. The study of these distances has allowed major progress on the geometry of spaces of probability measures, and has been used in many applications. It is therefore natural that the estimation of these distances has been a major topic of interest in the learning of optimal transport.

The key question here is the convergence in Wasserstein distance of an empirical distribution to the true distribution. Pioneering work on this topic began in the 80s and 90s (see, e.g., Ajtai et al., 1984; Talagrand, 1994), with the study of *Matching* (i.e. discrete optimal transport). Key statistical analysis of this problem includes finite sample bounds, see Horowitz & Karandikar (1994) and more recently Fournier & Guillin (2015); Weed & Bach (2019) among others, as well as distributional limits (see Tameling et al., 2019, and references therein).

Sadly, most work has remained limited to Wasserstein distances rather than generic cost functions, owing to a reliance on the pleasant geometric properties that they enjoy. This can be viewed as a special case of estimating the value of $\mathrm{Kant.}(\mu, \nu, c^*)$ when $(\mu, \nu)$ are unknown but $c^*$ is known and highly regular (a power of a distance).

**Estimation of entropic OT**    Motivated by the success of Entropic OT in designing numerical solution to OT problems, see Cuturi (2013), work on the Entropic problem has focused on estimating $\mathrm{Ent.}(\mu, \nu, c, \varepsilon)$ using $\mathrm{Ent.}(\hat{\mu}_n, \hat{\nu}_n, c, \varepsilon)$, for empirical measures $(\hat{\mu}_n, \hat{\nu}_n)$. This has often gone together with estimation for the Schrödinger potentials $(\varphi, \psi)$ of (20).

While this is very much the same type of study as for the Kantorovich problem in Wasserstein metrics, it should be noted

that the entropic problem exhibits qualitatively different behaviour. While learning the Kantorovich problem exhibits a curse of dimensionality, the entropic problem exhibits parametric-rate (dimension-free) convergence, as shown by (Genevay et al., 2019; Rigollet & Stromme, 2025). This was tempered by large dependencies in other problem quantities, which were reduced over time (Stromme, 2024) and complemented by distributional limits, see e.g. (Gonzalez-Sanz et al., 2024).

**Estimation of Monge maps**  While the estimation of Wasserstein distances is mostly motivated by statistical applications, the estimation of Monge maps is motived by effectively solving transport problems in an applied context. Here, one sees samples from two marginals $\mu$ and $\nu$, and attempts to estimate $T^*$ the minimiser of (40).

There has been a significant amount of machine learning and statistics literature on this topic, following on from (Hütter & Rigollet, 2021; Gunsilius, 2022). Various types of estimators have been constructed, either derived from optimal transport theory (Hütter & Rigollet, 2021), or from plug-in estimates using classical machine learning methods such as $k$-NN (Manole et al., 2024; Deb et al., 2021).

**Optimal transport applied to learning**  While these bibliographical notes concern learning in optimal transport, let us conclude by underlining that the machine learning community has used optimal transport to impressive success in applications. One could highlight in particular Wassertein GANs (Arjovsky et al., 2017) and subsequent works, e.g. by Salimans et al. (2018), as well as the field of domain adaptation (Courty et al., 2017; Torres et al., 2021).

**Online transport and matching**  When the transport problem itself must be solved online, i.e. $\mu$ and $c^*$ are known but $\nu$ is revealed sequentially and must be allocated immediately, the problem is known as *online transport* or *online matching* if $(\mu, \nu)$ are atomic. Applications of this can be found, e.g. in (Perrot et al., 2016; Alon et al., 2004; Johari et al., 2021; Min et al., 2022; Hatfield & Milgrom, 2005; Glorie et al., 2014). This problem is different from BOT both from a decision standpoint (allocated mass cannot be reallocated in the next round), and from an information standpoint (it is a combinatorial semi-bandit, see e.g. Jagadeesan et al. (2021); Sentenac et al. (2021)).

