# OpenReview forum: "Linear Bandits beyond Inner Product Spaces, the case of Bandit Optimal Transport"
_ICML.cc/2026/Conference — ICML 2026 regular_

### Official Review · Reviewer_bZD2 · 2026-03-02

**Soundness:** 3
**Presentation:** 3
**Significance:** 2
**Originality:** 2
**Overall Recommendation:** 3
**Confidence:** 3

**Summary:**

This paper studies a stochastic bandit problem, where the learner's objective is to solve find the Kantrovich's problem with bandit feedback for given measures $\mu, \nu$, which the authors call the BOT (bandit optimal transport) problem. Unlike the standard linear or kernel bandit problem, the objective function of this problem is not given as an inner product of a Hilbert space. Based on the fact that in the frequency domain the objective is given as an inner product of $L^2$ space under mild conditions, the authors construct a confidence set of the unknown cost function, and propose a UCB-type algorithm (EntUCB). Then, they provide a regret bound that depends on the actions of EntUCB. Assuming a basis of the $L^2$ space, they provide an approximation of EntUCB (basis^truncation EntUCB) and a regret bound that depends on the regularity of the cost function.

**Compliance With Llm Reviewing Policy:**

Affirmed.

**Final Justification:**

The author response partially resolves my concerns, but I still have a concern regarding motivation of the problem setting.

**Key Questions For Authors:**

- Could the authors provide motivating examples (or practical applications) for the problem setting?
- How to implement Algorithm 1 or 2? Is it possible to conduct numerical experiments?

**Limitations:**

Please refer to "Key Questions For Authors".

**Strengths And Weaknesses:**

### Strengths
- To the best of my knowledge, the results (methods and regret bounds) for the BOT (bandit optimal transport) problem are novel.
### Weaknesses
- The motivation for the problem setting (i.e., solving the Kantrovich problem with stochastic bandit feedback) is unclear.
- The proposed algorithm involves an optimization (Eq (11)) with respect to measures and functions, and the authors did not discuss implementation of the proposed method.
- The authors claim that the technical challenge of the BOT problem is that the objective is not given as an inner product of a Hilbert space, but in Section 3, they explain that the objective can be written as an inner product in the $L^2$ space after Fourier transformation under mild conditions, and fully utilize this fact for methods and regret analysis. So, the technical difficulties of the BOT problem are unclear.

---

> ### Author Rebuttal · Authors · 2026-03-27
>
> Dear reviewer,
>
> Thank you for your kind and careful attention to the paper and your detailed comments.
>
> ### Weaknesses
>
> > The motivation for the problem setting (i.e., solving the Kantrovich problem with stochastic bandit feedback) is unclear.
>
> See questions below.
>
> > The proposed algorithm involves an optimization (Eq (11)) with respect to measures and functions, and the authors did not discuss implementation of the proposed method.
>
> See questions below.
>
> > The authors claim that the technical challenge of the BOT problem is that the objective is not given as an inner product of a Hilbert space, but in Section 3, they explain that the objective can be written as an inner product in the space after Fourier transformation under mild conditions, and fully utilize this fact for methods and regret analysis. So, the technical difficulties of the BOT problem are unclear.
>
> In general, the objective *cannot* be written as an inner product; this is the core technical challenge. The algorithm is designed (via entropic penalisation of optimism in (11)) to *ensure* it only plays plans with the sufficient regularity conditions which ensure the inner product in (4) is valid. In other words, that equation (4) holds for every $\pi_t$ output of EntUCB is a consequence of its design, not a foregone fact. I will rewrite section 3.1 to better convey this point, as it is essential to understanding. I would like to apologise for the confusion this will have caused.
>
> ### Questions.
> Please find below discussions concerning your key questions.
> > Could the authors provide motivating examples (or practical applications) for the problem setting?
>    - I must stress that this is a new problem originating from theory, and thus the horizon of its practical applicability has not yet been explored. Nevertheless, I provided a possible example application in Appendix C.3. Let me recall it briefly for your convenience. Suppose you are a large industrial logistics firm that has to move large amounts of products between many suppliers and many purchasers. Every day, you have to decide how to route your supply trucks to deliver the goods at the lowest possible cost. For the sake of modelling, take a mean-field limit and represent the supply/demand/trucks by probability densities and assume that supply/demand is at a steady state (stationary). If, at the end of the day, you observe the day's transport cost, then your problem is exactly an instance of bandit optimal transport.
>
>    - As I've said, this is first and foremost a theory paper. From this perspective, the study of BOT is also motivated by other (semi-)bandit problems over measures, like reinforcement learning. I discuss this a bit on lines 1199--1240, should you be interested.
>
> - In regard to the value of infinite-dimensional bandits for bandit theory, please refer to the response to question 1 of reviewer 3 (fxE3) above.
>
> > How to implement Algorithm 1 or 2? Is it possible to conduct numerical experiments?
>  - Like most bandit papers which use optimistic methods, it is not obvious how to implement steps like the double maximisation in equation (11) efficiently. In some sense, this must be expected, as these methods are aiming to be optimally sample-efficient w.r.t. regret. Easy computation would be akin to a kind of free lunch.
>  - However, there are some specific problems with optimal transport that make even forming a coherent experimental format difficult. If you look only at discrete marginals $(\mu\_i)\_{i\in[n]}$ and $(\nu\_i)\_{i\in[n]}$, then the problem is finite-dimensional, and the whole phenomenon that the paper is focused on is invisible... If you start from continuous marginals $\mu$ $\nu$ and then solve some kind of approximation based on discretisations $(\mu\_i)\_{i\in[n]}$ and $(\nu\_i)\_{i\in[n]}$, you need to be able to recover a valid plan for $\mu$ and $\nu$ from them, otherwise you cannot compute the regret. Ignoring this issue can lead you to ``measure'' negative regret, which is obviously preposterous. See also the response to question 2 of Reviewer 1 (7avo). I will add a short discussion of computational issues to the paper.
> - Due to these theoretical challenges, an *experimentally sound* numerical exploration of this algorithm's performance is not currently within reach. I believe that, if there is nothing sound to say, it is best to stay silent and thus haven't proposed any experiments which would be inherently unsound.
>
> I hope this reply addresses all your concerns, allowing you to revise your review. If not, please feel free to ask any further questions.
>
> Thank you for your consideration.
>
> Kind regards.

---

> > ### Author Rebuttal · Reviewer_bZD2 · 2026-04-03
> >
> > I appreciate the detailed responses. In particular, my question about the inner product in a Hilbert space has been resolved. However, I still have concerns regarding realistic applications and implementation of the algorithm. Therefore, I will maintain the my score.

---

### Official Review · Reviewer_fxE3 · 2026-03-11

**Soundness:** 3
**Presentation:** 4
**Significance:** 2
**Originality:** 3
**Overall Recommendation:** 4
**Confidence:** 3

**Summary:**

The setting of the usual linear bandit problem is that in each period $t$, a player gets noisy feedback $C_t = \langle a_t, \varsigma\rangle + \xi_t$ when using action $a_t$, and having to choose future actions such as to minimize regret (the summed differences $C_t - \inf_a \langle a, \varsigma \rangle)$. In particular, this _linear_ problem (opposed to general non-linear cost structures) is the only setting being tractable in infinite dimensions so far, and the paper notes that even the term "linear" is overstating the generality of the setting, as the theory so far is limited to inner product costs (i.e., necessitates a Hilbert space structure). To extend this, the paper introduces a new linear setting which falls outside of this regime, namely the bandit optimal transport problem, where $C_t = \int c d\pi_t + \xi_t$, so where actions are couplings $\pi_t \in \Pi(\mu, \nu)$ and the real cost function $c$ is unknown. The basic idea is to show that the cost term $\int c d\pi_t$ can be _approximated_ by an inner product in a Hilbert space (of Fourier coefficients of both $c$ and $\pi_t$), at least for suitably smooth $\pi_t$. Hereby, the restriction of $\pi_t$ being sufficiently smooth is not required as an a-priori assumption, but it is shown that the player can simply choose such $\pi_t$ via solutions to a suitable regularized (entropic) optimal transport problem. Regret bounds are shown which are in line with existing theory for inner-product bandit problems.

**Compliance With Llm Reviewing Policy:**

Affirmed.

**Final Justification:**

Overall I think it is a clearly written paper which achieves its goal. While the rebuttal outlined some potential for impact in the future to address my main concern regarding limited significance, I still think the setting of the paper is fairly specific and will probably not immediately lead to broad impact (even though I see the point in the rebuttal that the introduced methods can generalize to different settings), hence I keep my recommendation of "weak accept".

**Key Questions For Authors:**

Mostly mentioned above, to summarize
1) Can you elaborate on the importance of infinite-dimensional settings for bandit problems? In which sense is the studied setting or the explored methods relevant with respect to this general importance?
2) Is there hope to calculate a real example in this setting showcasing the results? Can perhaps even a general method involving increasingly good approximations by discrete marginals be formulated?
3) Would the $\varepsilon \log(1/\varepsilon)$ versus $\varepsilon$ difference for the approximation by regularized OT make a difference?

**Limitations:**

yes

**Strengths And Weaknesses:**

Strengths:
- The problem was explained clearly and the goals, contributions and limitations communicated precisely. I also found the presentation of the results very suitable and relatively (relative to their complexity) easy to follow.
- I think the general aspect of "only a subset of actions are tractable" (in the paper, those with $L^2$ densities), but "playing only this subset of actions can approximate arbitrary actions sufficiently well" (in the paper, the required approximation results for regularized OT) is very important, and the paper makes a good first step in this direction. So while the basic strategic approach (how to choose actions and assess costs) is very similar to the Hilbert space setting, I think the methodology to work with this approximation strategy was quite original.
- The paper does what it claims, there are no hidden strong assumptions (Assumptions 3.1 and 3.2 are surprisingly weak) and as far as I can tell, the math is correct.

Weaknesses:
- The paper is of course limited in scope. The studied setting is only one among many linear bandits which are not inner product costs, so the significance is limited. Further, the importance of infinite dimensional settings for the bandit problem could be explored a bit more in the introduction.
- I found the example applications (in the appendix) interesting in terms of their description, but relatively limited in terms of how they were explored. Analytically calculating a real example or numerically illustrating an example would have added a lot to the paper in my opinion.
- Of course, currently the paper does not explore at all how these infinite-dimensional settings can be made tractable numerically. But I am wondering: Is it so much more difficult to add an increasingly fine approximation by discrete marginals $(\mu_t, \nu_t)$, similarly to how currently the paper uses an increasingly small penalization parameter $\varepsilon_t$? It would only lead to another (arbitrarily small) error term in the feedback, but it may yield practical implications (what kind of discrete approximation quality is necessary for optimal regret).

General comment:
- Currently the paper uses the result for the approximation of normal OT by entropic OT of order $\varepsilon \log(1/\varepsilon)$, which is an approximation _including_ the entropy term. But for the setting of the paper, the approximation without entropy term should be sufficient, which is actually of order $\varepsilon$ (at least perhaps under slightly stronger assumptions on the cost, see "Convergence rates of the regularized optimal transport: disentangling suboptimality and entropy" by Malamut and Sylvestre, 2025). Would this improvement from $\varepsilon \log(1/\varepsilon)$ to $\varepsilon$ change anything?

Small comments:
- Page 2 states "..., which is a linear functional on the space of probability measures", which is misleading, as only the underlying cost of the OT problem ($\pi \mapsto \int c d\pi$), not the whole OT problem ($(\mu, \nu) \mapsto OT(\mu, \nu)$) is linear.
- At the start of page 4, I guess it should be $c^*$ as well when defining $C_t$?
- As far as I can see, the main text currently never mentions how $\varrho$ is chosen, which is confusing.
- After Assumption 3.2, I think it should be $(\pi_s)_{s \leq t}$ (and same for $C_t$).

---

> ### Author Rebuttal · Authors · 2026-03-27
>
> Dear reviewer,
>
> Thank you for your kind and careful attention to the paper and your detailed comments.
>
> Please find below responses to your main questions, as numbered.
>
> > 1.  Can you elaborate on the importance of infinite-dimensional settings for bandit problems? In which sense is the studied setting or the explored methods relevant with respect to this general importance?
> - Let me be clear: as of today, the interest in infinite-dimensional bandits is almost completely theoretical.
> While I believe there is potential for practical applications (indeed, optimal transport has found real applications), my concern in infinite dimensions is to give a clear and rigorous picture of what is and isn't learnable under bandit feedback.
> - In this vein, infinite-dimensional bandits have been the main driver of general progress. While most bandit works study specific finite-dimensional settings, the shortcoming of this approach is that it doesn’t lead to the emergence of broad *classes* of learnable objectives, just isolated islands of learnability. In contrast, a Kernel bandit for Matern kernels tells you that any function in this broad class (whose analytical properties are well known) is learnable, at what rate, and provides the algorithm to do it. This ability to obtain general transversal results is the value of these infinite-dimensional formulations. However, these works typically remain limited to inner-product (Kernel) bandits.
> - The relevance of this work is in overcoming this limitation of infinite-dimensional bandits by taking a first step outside of the inner product world. By showing that linear functionals on non-Hilbert spaces can be learned under some regularity conditions (see Appendix D.2, which goes beyond OT), it opens the way to a broader theory of bandit learnability, which will carry the benefits of the infinite-dimensional approach to new problems, such as the non-linear Monge problem (see Appendix G.2).
>
>
> > 2. Is there hope to calculate a real example in this setting, showcasing the results? Can perhaps even a general method involving increasingly good approximations by discrete marginals be formulated?
> - I have thought about this question quite a bit and, unfortunately, it's quite complicated.
> - The core problem with discretisation is in the regret analysis. Suppose you have discretised $\mu$ and $\nu$ onto $(x\_i)\_{i\in[n]}$ and $(y\_i)\_{i\in[n]}$, yielding $(\mu\_i)\_{i\in[n]}$ and $(\nu_i)\_{i\in[n]}$ and that you solve the corresponding discrete Entropic OT problem perfectly (remember that Sinkhorn outputs do not satify the constraints). You can’t just insert this solution into the continuous OT functional because it doesn’t satisfy the continuous marginal constraints; if you ignore this, you can get negative regret. So you need a way to ``de-discretise a coupling’’, and it needs to be explicit enough to be able to integrate against the resulting coupling of $(\mu,\nu)$.
> - Because of several problems like this one, many technically intricate building blocks from a range of fields are required to give a rigorous and experimentally sound showcase of EntUCB. Until all the pieces are in place, it will have to remain a largely theoretical endeavour.
>
>
> > 3. Would the $\varepsilon \log(1/\varepsilon)$ versus $\varepsilon$ difference for the approximation by regularised OT make a difference?
> - It will not make a difference in the regret as the approximation term (whether $\varepsilon \log(1/\varepsilon)$ or $\varepsilon$) ends up as a lower-order term anyway. This is why I chose to present only the simplest result from the paper by Carlier *et al*. However, the study of these approximation results is an exciting new direction, so thank you for bringing this recent paper to my attention.
>
>
> I hope this reply addresses all your concerns, allowing you to revise your review. If not, please feel free to ask any further questions.
>
> Thank you for your consideration.
>
> Kind regards.

---

> > ### Author Rebuttal · Reviewer_fxE3 · 2026-04-01
> >
> > Just one side remark: De-discretizing a coupling is reasonably possible, as quantization approximation by discrete marginals via Voronoi partitions leads fairly tractable joint distributions between marginals and their discretization, and then using the shadow coupling construction as for instance used in [1] may lead to the desired de-discretized couplings. But that is perhaps beyond the scope of the current work.
> >
> > In any case, I appreciate your clear reply. I still think it's a nice paper with somewhat limited significance, so I will keep my initial score.
> >
> > [1] Eckstein, S., and Nutz, M. "Quantitative stability of regularized optimal transport and convergence of Sinkhorn's algorithm." SIAM Journal on Mathematical Analysis 54.6 (2022): 5922-5948.

---

### Official Review · Reviewer_r4qz · 2026-03-12

**Soundness:** 3
**Presentation:** 4
**Significance:** 2
**Originality:** 2
**Overall Recommendation:** 4
**Confidence:** 4

**Summary:**

The paper considers the problem of *linear* bandits, where the authors aim to focus on *linear* functions that not formulated as an inner product. They study the specific example of bandit optimal transport where the actions are the transport plan and there is an underlying cost vector for which the learner seeks the optimal transport plan. The authors show how classical approaches like OFUL can be used for this problem. In particular, they focus on the problem of entropy regularized BOT (equivalent to l2 regularized bandits), propose an OFUL based algorithm and provide its regret analysis.

**Compliance With Llm Reviewing Policy:**

Affirmed.

**Final Justification:**

The author's rebuttal helped me better understand some aspects of the paper and I have updated my score to reflect that.

**Key Questions For Authors:**

I have the following questions.

1. Why do you need the Fourier transform step? Firstly, based on assumption 3.1, $c^* \in L^2$ (I am assuming the measure is always $\varrho$, as in the paper). This automatically ensures that $c^* $ is in a Hilbert space. Then in Eqn 4, you go on and eventually assume that the density $d\pi /d\varrho \in L^2$.  At this point, the integral $\int c^* \ d\pi = \int c^* \ (d\pi /d\varrho) d\varrho = \langle c^*, (d\pi /d\varrho)  \rangle_{L^2}$, we already have an inner product formulation. Had the inner product only been valid in Fourier space, I would have understood the need for the step. But after all assumptions, I don't see the need for that extra step. Am I missing something?

2. If my understanding is correct, it raises another question. In problem formulation section, the authors claim:
> This is a bilinear form reminiscent of (2), except for the fact that the bracket is not an inner product. Due to this (because $\pi$ and $c$ do not belong to a common Hilbert space) there are neither Riesz representation theorem nor Cauchy-Schwarz inequality that apply.

I agree with this statement in the general case. The entire motivation of the paper is that the reward function is *linear but not inner product*. As explained above, unlike the general case, they now belong to a common Hilbert space and the function is again an inner product! This is in straight contradiction with what is claimed in the abstract
> we show that an inner product structure is *not* necessary to achieve efficient learning in linear bandits. We propose a refinement of the classical OFUL algorithm that operates by embedding the action set into a Hilbertian sub-space, where confidence sets can be built via least-squares estimation.

There is no *embedding* (at least no need for one), you end up *assuming* the Hilbertian subspace part. This is a very big distinction.
The embedding via Fourier route is not necessary. The approach and paper are still interesting, however, they sort defeat the entire motivation at the beginning of the paper.

3. Isn't the Entropic Optimal Transport problem (as stated in Eqn (10)) identical to the contextual bandit problem with KL regularization (see [1] for an example reference). The KL regularized problem has a closed form optimal solution. Why can't we directly use it here?

[1] Heyang Zhao, Chenlu Ye, Quanquan Gu, Tong Zhang, Sharp Analysis for KL-Regularized Contextual Bandits and RLHF

**Limitations:**

Yes.

**Strengths And Weaknesses:**

Strengths: The paper is nicely written. It is a welcome change to see a paper so thoroughly and clearly written. Also, I like the key approach used in the paper. It is crisp and has potential other applications and the analysis seems sound as well.

I don't see any glaring weakness per se, but do I have some questions that I have documented in the next section.

---

> ### Author Rebuttal · Authors · 2026-03-27
>
> Dear reviewer,
>
> Thank you for your kind and careful attention to the paper and your comments on the writing, which are highly appreciated.
>
> Please find answers to your questions below, numbered as in your review. Notice that I have reordered two of them for clarity.
>
> Q2. I want to make abundantly clear that the entire challenge comes from the fact that we *absolutely cannot* assume that an optimal plan $\pi^*$ has a density (let alone one in $L^2(\mathbb{R}^d,\varrho)$). (In fact, this is almost never the case if $\mu$ has a density w.r.t. the Lebesgue measure (See, e.g., Villani 2009, Ch. 9--10).) In this work, I *do not* assume that any plan has density; instead, I construct an algorithm that only ever plays plans $\pi_t$ with a density by penalising optimism by plan entropy (see the discussion in Section 3.3). This algorithmic adaptation ensures that the RLS confidence sets in $L^2(\mathbb{C}^d,\varrho)$ remain well-defined and sets the algorithm apart from OFUL.
>
> Q1. First, I would like to apologise for the confusion the writing surrounding equation (4) has caused. This equation was meant to be only a formal calculation (i.e. which holds only when its terms are well-defined, which is to say when $\mathrm{d}\pi/\mathrm{d}\varrho\in L^2(\mathbb{R},\varrho)$) and not to suggest that I assume $\mathrm{d}\pi/\mathrm{d}\varrho\in L^2(\mathbb{R},\varrho)$. As stated in Q2 above, that is the opposite of what I want.
>
> Second, regarding the topic of the Fourier transform. As you are doubtless aware (see also l203, right), $F\pi\in L^2(\mathbb{R},\varrho)$ is equivalent to $\mathrm{d}\pi/\mathrm{d}\varrho\in L^2(\mathbb{R},\varrho)$, and $F$ is an isometry so one can also form an inner product without it under the same conditions, which you give in your question. I chose to present the frequency domain form directly because I think that it gives a better intuition of paths to generalise to other linear functionals, which is done in Appendix D.2 (l1090). Therein is also where the "embedding" terminology is used to its full extent.
> I think of this paper as a first step in the direction of settling the ``linear vs inner product’’ question, which is why I care about generalisations, but I would like to avoid confusion, so I will revise Section 3.1 a bit to make this explicit.
>
> I hope that these answers to both Q1 and Q2 have clarified that the work's assumptions are, in fact, in line with the motivation and that the issue of the lack of inner product structure for Least-squares is addressed by a contribution in algorithmic design and not by assumption.
>
> Q3. In short, no, the two settings are quite different. (If you are interested, please find below a detailed list of differences.) However, the two works are conceptually close, and I was not aware of this line of work, so thank you for bringing it to my attention. I will add a discussion of this line of work, especially as to how it relates closely to what is discussed in lines 1202-1240.
> - The main difference is that the problem of Zhao *et al.* is unconstrained, while Entropic OT is still constrained by the marginals. This means their closed-form solution is not the correct one for Entropic OT; it is missing an $e^{\phi\oplus\psi}$ factor, where $(\phi,\psi)$ are the Kantorovich dual potentials.
> - They make a coverage assumption by a reference policy, which removes the need for sharp balancing of exploration and exploitation.
> - When they consider regret, they are taking regret relative to the entropic functional, not the deterministic one like us. This avoids the need for approximation results such as this paper’s Lemma C.3. In general, this is a major difficulty, as highlighted in Appendix D.2.
> - They are working with a generic reward, not necessarily a bilinear form, which is why they have to use covering arguments.
> - Technically, actions are discrete in their setting, but I think their work should generalise *mutatis mutandis*.
>
>
> I hope this reply addresses all your concerns, allowing you to revise your review. If not, please feel free to ask any further questions.
>
> Thank you for your consideration.
>
> Kind regards.

---

> > ### Author Rebuttal · Reviewer_r4qz · 2026-04-03
> >
> > I absolutely agree that any assumption on $\pi^{\star}$ simplifies the problem. I think I got confused with the wording. So effectively the setup is as follows: $c^{\star} \in L^2$, no assumption on $\pi^{\star}$. You consider the entropic optimal transport, whose error to the optimal transport is well documented. Now, while designing the algorithm, you focus only on policies whose L2 norm is finite.You establish that optimal policies for EOT are infact in L2, which is why it is sufficient to consider this subclass. I am still a bit unsure of the need to go via Fourier path.
> >
> > The KL regularized contextual bandits might have several good relations to your problem. Since this problem has received a lot of attention even in practical settings, the relation might help motivate your work more, especially in the light of practical considerations. I want strongly recommend you to look through in more detail (I had just shared one representative paper; the literature is actually quite large). I feel the marginal constraint is the most likely to be the major point of difference.

---

### Official Review · Reviewer_7avo · 2026-03-12

**Soundness:** 4
**Presentation:** 3
**Significance:** 4
**Originality:** 4
**Overall Recommendation:** 6
**Confidence:** 3

**Summary:**

This paper highlights the theoretic gap between linear bandits and their prevalent subset of inner product bandits. Optimal transport is framed as a bandit problem that exemplifies this complementary realm. To solve the task by sequential learning, an entropy regularised UCB algorithm with confidence sets in the frequency domain of the measures is developed. Both trajectorial and, further integrating basis decomposition and truncation into the algorithm, worst-case regret bounds are established. The authors also offer discussions on the background and research extensions.

**Compliance With Llm Reviewing Policy:**

Affirmed.

**Key Questions For Authors:**

1. Are there ways to demonstrate the effectiveness of this bandit approach over other optimal transport algorithms?

    2. For which portions of the algorithms can approximate computational complexity expectations be provided?

**Limitations:**

Yes

**Strengths And Weaknesses:**

Strengths:
    - Contributions are novel and significant towards filling the gap identified for linear bandits with infinite dimensional action spaces.
    - The main theoretical concepts and tools utilised in the design process are well motivated and elaborated upon.
    - Assumptions seem reasonable and minimalistic.
    - Notation and style are quite meticulous.

Weaknesses:
    - It is uncertain whether the optimisation step of the algorithms can even be implemented numerically. As such, experiments are not available.
    - Abstract and conclusion are in parts overly detailed recapitulations.
    - Very minor: Thm. 4.1 should specify what sort of quantity $\kappa$ is. The join and meet operators from order theory could be named in the notation section.

---

> ### Author Rebuttal · Authors · 2026-03-27
>
> Dear reviewer,
>
> Thank you for your kind and careful attention to the paper. In particular, thank you for pointing out the missing specification of the constant $\kappa=\kappa(\mu,\nu,\alpha)\in\mathbb{R}$, which will be corrected.  On your advice, I will also revise the abstract and conclusion.
>
> In regard to your questions, please find answers below:
>
> > 1. Are there ways to demonstrate the effectiveness of this bandit approach over other optimal transport algorithms?
> - It is important to clearly distinguish the optimal transport (OT) problem, which is a static optimisation problem, from the bandit OT problem, which is a sequential *learning* problem. All optimisation algorithms for OT (Sinkhorn, Monge-Ampère solvers, etc.) require knowledge of the cost function, so comparing EntUCB to them is akin to regret (i.e. cumulated cost of uncertainty about $c^*$). Thus, regret bounds like those in the paper are the best way to understand the relative effectiveness of EntUCB relative to an OT solver.
>
> > 2. For which portions of the algorithms can approximate computational complexity expectations be provided?
>
> It really depends on $\mu$ and $\nu$.
> -  If they are both discrete, say $\lvert \text{supp}(\mu)\rvert=\lvert \text{supp}(\nu)\rvert=n$, then the regularised least-squares is solvable via SVD in $\mathcal{O}(n^3)$ (or faster), while an $\epsilon$-solution to the entropic problem for $c^*\in\mathbb{R}^{n\times n}$ fixed can be obtained via Sinkhorn’s algorithm in $\mathcal{O}(n^2\log(n/\epsilon^2))$. A naive optimisation of the optimism step (i.e., the inner minimum in (10)) can be done by simply taking the smallest permissible entry for each entry $(i,j)$, so $\mathcal{O}(n^2)$.
>
> - If one wants to handle continuous measures, I have detailed how to implement least-squares with the method of Section 5. Under infinite memory, Sinkhorn continues to converge linearly, so we can solve one of the two minima in (11). However, optimism seems difficult to implement even with infinite memory. I am not aware of any work on the approximate resolution of this kind of problem, either. Having thought about the scale of the task, I think it is achievable but still pretty far away.
>
> - If one wants to implement with finite memory, the problem starts to fray at the edges, and measure-theoretic issues appear all over the place. Let me give only one example. Finite memory requires discretising the measures $\mu\to(\mu\_i)\_{i\in[n]}$ and $\nu\to(\nu\_i)\_{i\in[n]}$ at some point, if only to be able to store past plans $\pi_t$ in memory for least-squares. But then one needs a way to reconstruct a valid plan for $(\mu,\nu)$ from the solution of the problem with discretised marginals, a difficult problem which is well beyond the learning theory scope of this work.
>
> Overall, I think you are right to underline this as an underaddressed point in the manuscript. I will add a short discussion of these considerations to the paper.
>
> Thank you for your consideration. Please feel free to ask any further questions.
>
> Kind regards.

---

> > ### Author Rebuttal · Reviewer_7avo · 2026-04-03
> >
> > Many thanks for providing additional elaborations. The comparison in question 1 was intended towards online optimal transport algorithms, but I see Appendix H also discusses some key differences regarding feedback and decision permanence to the bandit formulation here.

---

### Decision · Program_Chairs · 2026-04-30

**Decision:**

Accept (regular)

**Comment:**

Strength:
- The paper extends the problem of linear bandits to infinite-dimensional spaces and aims to go beyond "linear = inner product" bandits (although the solution still reduces to an inner product).
- The research question of extending linear bandits beyond inner products is interesting, the paper is well-written, and the theoretical contribution is solid.

Weakness:
- The proposed algorithm has no computationally efficient implementation.
- The practical motivation of the proposed infinite-dimensional setting is unclear (even after the rebuttal).
- Since the paper starts with looking at "linear, but not inner product" setting, but then solves it using an inner product approximation, the message of the paper could probably be revised to reflect this.

Overall, this is an interesting theoretical research direction (although with no obvious practical applications yet) that was positively received by the reviewers. I recommend acceptance.